# GSTA4 mediates reduction of cisplatin ototoxicity in female mice

Hyo-Jin Park[1,9], Mi-Jung Kim [1,9], Christina Rothenberger [1,9], Ashok Kumar [2], Edith M. Sampson [3], Dalian Ding[4], Chul Han[1], Karessa White [1], Kevin Boyd [1], Senthilvelan Manohar [4], Yong-Hwan Kim[5], Maria S. Ticsa[1], Aaron S. Gomez [1], Isabela Caicedo[1], Upal Bose [1], Paul J. Linser[6], Takuya Miyakawa [7], Masaru Tanokura [7], Thomas C. Foster[2], Richard Salvi [4,8] & Shinichi Someya [1]

Cisplatin is one of the most widely used chemotherapeutic drugs for the treatment of cancer. Unfortunately, one of its major side effects is permanent hearing loss. Here, we show that glutathione transferase α4 (GSTA4), a member of the Phase II detoxifying enzyme super-family, mediates reduction of cisplatin ototoxicity by removing 4-hydroxynonenal (4-HNE) in the inner ears of female mice. Under cisplatin treatment, loss of *Gsta4* results in more profound hearing loss in female mice compared to male mice. Cisplatin stimulates GSTA4 activity in the inner ear of female wild-type, but not male wild-type mice. In female *Gsta4*[−/−] mice, cisplatin treatment results in increased levels of 4-HNE in cochlear neurons compared to male *Gsta4*[−/−] mice. In CBA/CaJ mice, ovariectomy decreases mRNA expression of *Gsta4*, and the levels of GSTA4 protein in the inner ears. Thus, our findings suggest that GSTA4-dependent detoxification may play a role in estrogen-mediated neuroprotection.

[1] Department of Aging and Geriatric Research, University of Florida, Gainesville, FL 32611, USA. [2] Department of Neuroscience, University of Florida, Gainesville, FL 32611, USA. [3] Monoclonal Antibody Core, Interdisciplinary Center for Biotechnology Research, University of Florida, Gainesville, FL 32610, USA. [4] Center for Hearing and Deafness, State University of New York at Buffalo, Buffalo, NY 14214, USA. [5] Department of Neurobiology, Barrow Neurological Institute, Phoenix, AZ 85013, USA. [6] Whitney Laboratory, University of Florida, St. Augustine, FL 32080, USA. [7] Department of Applied Biological Chemistry, University of Tokyo, Yayoi, Tokyo 113-8657, Japan. [8] Department of Audiology and Speech-Language Pathology, Asia University, Taichung, Taiwan 41354, Republic of China. [9] These authors contributed equally: Hyo-Jin Park, Mi-Jung Kim, and Christina Rothenberger. Correspondence and requests for materials should be addressed to S.S. (email: someya@ufl.edu)

Cells possess a wide range of detoxification enzymes capable of removing thousands of naturally occurring toxins and foreign chemicals[1–3]. The metabolism of those toxic compounds is generally divided into phase I and phase II reactions. Most phase I enzymes are capable of both detoxification and metabolic activation, and cytochrome P450 enzymes comprise 70–80% of all phase I enzymes[4]. Phase II enzymes catalyze conjugation reactions, producing a less toxic, hydrophilic product that can be readily excreted from the cell through transmembrane transporters, completing the detoxification cycle[1–3,5]. Glutathione transferases (GSTs) are a superfamily of phase II detoxifying enzymes that protect cells by catalyzing conjugation reactions of toxic compounds to reduced glutathione (GSH). Over 20 mammalian GSTs have been identified to date. GSTs can be divided into three major families: cytosolic, mitochondrial, and membrane-bound microsomal[1,2]. Of these, cytosolic GSTs constitute the largest family[1,3,5]. The most well-characterized cytosolic GST classes have been named alpha (GSTA), mu (GSTM), pi (GSTP), and theta (GSTT). Cytosolic GSTs are dimeric, with subunit molecular weights of approximately 25 kDa. Each subunit contains a catalytically independent active site that consists of a GSH-binding site in the amino-terminal domain, and a site that binds the xenobiotic substrate in the carboxy-terminal domain[2]. Most cytosolic GST classes show a high degree of polymorphism[3].

The alpha-class GSTs consist of 5 distinct members, GSTA1, GSTA2, GSTA3, GSTA4, and GSTA5[1,5,6]. GSTA1 and GSTA2 possess antioxidant activity toward peroxides, fatty acids, and hydroperoxides[6,7], while GSTA4 possesses high catalytic efficiency toward 4-hydroxynonenal (4-HNE)[8–11], a cytotoxic end product of lipid peroxidation that covalently modifies protein and DNA and contributes to neurodegenerative diseases, aging, and cancer[8,9,12]. Conjugation to GSH by GSTA4 is thought to be a major route of 4-HNE elimination. In support of this idea, knockdown of *Caenorhabditis elegans gst-10* that encodes a detoxifying enzyme with high catalytic activity toward 4-HNE, reduced lifespan[13], while overexpression of *gst-10* or murine *Gsta4* increased lifespan[14]. In human erythroleukemia cell lines, overexpression of *GSTA4* decreased the levels of 4-HNE[15] and protected cells against $H_2O_2$ (hydrogen peroxide) or UV-A (ultraviolet A) mediated apoptosis[12]. Mice deficient for *Gsta4* are viable and appear phenotypically normal, but displayed increased susceptibility for bacterial infection[1,16]. Under treatment with paraquat, a toxic herbicide known to induce oxidative stress and cell death, *Gsta4* null mice exhibited reduced survival compared to wild-type (WT) mice. These reports indicate that GSTA4 plays an important role in aging and cellular survival under oxidative stress conditions.

Cisplatin is one of the most widely used chemotherapeutic agents for the treatment of a broad spectrum of cancers[17–19]. However, cisplatin chemotherapy commonly causes permanent hearing loss in 40–80% of patients of all ages. Hearing loss has lifelong ramifications, particularly leading to difficulty with language acquisition and social isolation in young children. Cisplatin-induced hearing loss is dose-dependent, irreversible, and associated with loss of cochlear hair cells (HCs). Cisplatin is thought to exert its cytotoxic effects through DNA cross-linking and generation of reactive oxygen species following binding to cytoplasmic proteins, leading to increased oxidative damage and cell death[17,20]. Cisplatin also increases the levels of 4-HNE and malondialdehyde, another end product of lipid peroxidation, and decreased levels of GSH in the kidney[21,22]. In the cochlea, the sensory organ of hearing, cisplatin-induced ototoxicity is thought to be initiated by its uptake into the sensory HCs (HCs), spiral ganglion neurons (SGNs), and/or stria vascularis (SV) cells[17,23]. Cisplatin administration also results in increased levels of 4-HNE in rat outer HCs[24]. However, much of our understanding of GST detoxification function comes from studies using livers and kidneys, and it is unclear if GST detoxifying enzymes play a significant role in cisplatin ototoxicity.

In the current study, we show that GSTA4 mediates reduction of cisplatin ototoxicity by removing 4-HNE in the inner ears of female mice. Under cisplatin treatment, loss of *Gsta4* results in more profound hearing loss in female mice compared to male mice. Cisplatin stimulates GSTA4 activity in the inner ear of female WT, but not male WT mice. In female *Gsta4*[−/−] mice, cisplatin treatment results in increased levels of 4-HNE in cochlear neurons compared to male *Gsta4*[−/−] mice. In CBA/CaJ mice, females show increased mRNA expression of *Nrf2*, a regulator of Phase II detoxification genes, and 9 genes involved in Phase II detoxification, including *Gsta4*, compared to males. In contrast, ovariectomy decreases mRNA expression of *Nrf2* and 25 genes involved in Phase II detoxification, including *Gsta4*, and the levels of GSTA4 protein in the inner ears. Thus, our findings suggest that GSTA4-dependent detoxification may play a role in estrogen-mediated neuroprotection.

## Results

**Backcrossing *Gsta4*[+/−] mice onto the CBA/CaJ mouse strain.** Heterozygous *Gsta4*[+/−] mice were backcrossed for six generations onto the CBA/CaJ mouse strain, a normal-hearing strain that does not carry the early onset hearing loss susceptibility allele (*Cdh23*[753A])[25,26]. The CBA/CaJ strain is considered a model of sex differences in hearing and longevity because female CBA mice lose hearing more slowly than males[27] and live longer than males[28]. CBA mice also maintain normal auditory function until late in life[25–27]. We genotyped N6 WT (CBA/CaJ-*Gsta4*[+/+] or WT) and homozygous (CBA/CaJ-*Gsta4*[−/−] or *Gsta4*[−/−]) mice by polymerase chain reaction (PCR) genotyping (Supplementary Fig. 1a) and then sequenced the *Cdh23* (cadherin 23) gene in the DNA obtained from the tails of these mice. We confirmed that all WT and *Gsta4*[−/−] mice had the same WT *Cdh23* genotype (*Cdh23*[753G/753G]) (Supplementary Fig. 1b). Young mice lacking the *Gsta4* gene on the CBA/CaJ background appeared phenotypically normal and no significant differences were observed in body weight between male or female *Gsta4*[+/+] and *Gsta4*[−/−] mice (Supplementary Fig. 2). These observations are consistent with the phenotypic data of *Gsta4* homozygous mice on the 129S5/SvEvBrd;C57BL/6J background reported by Lexicon Genetics[29].

**Localization of GSTA4 in mouse cochlea.** To confirm that GSTA4 protein is expressed in the inner ear of WT mice and to validate the genotyping results, we first measured GSTA4 protein levels in the cytosol of inner ear tissues from 5-month-old male WT and *Gsta4*[−/−] mice by Western blotting. *Gsta4*[−/−] mice displayed no detectable level of GSTA4 protein in the inner ear (Fig. 1a). Next, we investigated the subcellular-localization of GSTA4 in the cochlea of 5-month-old male WT mice. GSTA4 was immunostained with anti-GSTA4 antibody and observed by confocal microscopy. Figure 1b–d shows an area of the organ of Corti (OC) in the cochlea at low magnification: GSTA4 immunostaining was prominent in the SV (Fig. 1k–m). GSTA4 immunostaining was also present in SGNs (Fig. 1e–g) and the OC (Fig. 1h–j). In contrast to WT mice, GSTA4 immunolabeling was completely absent from all *Gsta4*[−/−] cochlear tissues as illustrated by images from the OC (Fig. 1n–p).

**Cisplatin causes more profound hearing loss in female *Gsta4*[−/−] mice.** To investigate the role of GSTA4 in cisplatin ototoxicity in mice, we first evaluated the three-cycle cisplatin administration protocol developed by Lisa Cunningham and colleagues[19,23] using female CBA/CaJ mice. Young female CBA/

CaJ mice received 2–3 cycles of cisplatin treatment at 3.0 or 3.5 mg/kg/day. To assess hearing sensitivity in CBA/CaJ mice prior to and after cisplatin administration, we measured auditory brainstem response (ABR) thresholds with a tone burst stimulus over a broad frequency range (4–64 kHz). At 3.5 mg/kg/day, three-cycle cisplatin resulted in profound hearing loss across all frequencies, while two-cycle cisplatin resulted in severe to profound hearing loss at the high frequencies (Supplementary Fig. 3a). The three-cycle cisplatin administration also caused severe body weight loss: the body weight of cisplatin-treated mice decreased by ~20%, and importantly the animals failed to gain body weight or their body weight did not rise to above 80% of the original weight after the second 10-day recovery period (Supplementary Fig. 3b). At 3.0 mg/kg/day, three-cycle cisplatin also resulted in severe to profound hearing loss at the high frequencies, while two-cycles of cisplatin resulted in moderate hearing loss at the high frequencies. Both two- and three-cycle cisplatin administration at 3.0 mg/kg/day resulted in moderate weight loss: the body weight of cisplatin-treated mice decreased by approximately 10%, but the animals were able to gain body weight or their body weight rose above 95% of the original weight at the end of second or third 10-day recovery period (Supplementary Fig. 3b). Because we wished to: (1) determine if there were any differences in hearing sensitivity between WT and $Gsta4^{-/-}$ mice under cisplatin treatment and (2) develop a cisplatin administration protocol where WT mice would display moderate hearing loss and moderate weight loss to keep the mice healthy, male and female WT and $Gsta4^{-/-}$ mice were treated with the two-cycle cisplatin administration protocol (3.0 mg/kg/day): mice received 3 mg/kg of cisplatin each day for 4 days. The 4-day cisplatin administration period was followed by 10 days of recovery. This 14-day protocol was repeated once more for a total of two cycles of cisplatin administration.

To assess hearing sensitivity in male and female WT and $Gsta4^{-/-}$ mice prior to and after cisplatin administration, we measured ABR thresholds with a tone burst stimulus over a broad frequency range in WT and $Gsta4^{-/-}$ mice. In mice that did not receive cisplatin administration (precisplatin), mean ABR thresholds in both male and female WT and $Gsta4^{-/-}$ mice ranged from approximately 24–42 dB SPL (SPL) at the low to high frequencies (8–48 kHz), indicating normal hearing (Fig. 2a, b). There were no differences in ABR thresholds at all frequencies between male WT and $Gsta4^{-/-}$ mice or between female WT and $Gsta4^{-/-}$ mice. As expected, the two cycle cisplatin protocol at 3.0 mg/kg/day resulted in a moderate weight loss in both male and female WT and $Gsta4^{-/-}$ mice (Fig. 2c), but there were no differences in body weight between male WT and $Gsta4^{-/-}$ mice or female WT and $Gsta4^{-/-}$ mice. In cisplatin-treated male mice, mean ABR thresholds in both WT and $Gsta4^{-/-}$ mice ranged from approximately 28–83 dB SPL at the low to high frequencies (4–64 kHz) (Fig. 2a, b). There were no differences in ABR thresholds at all frequencies between male WT and $Gsta4^{-/-}$ mice. In cisplatin-treated female mice, mean ABR thresholds in both WT and $Gsta4^{-/-}$ mice ranged from approximately 26–88 dB SPL and from 4–64 kHz. However, in general, ABR thresholds were substantially higher in cisplatin-treated female $Gsta4^{-/-}$ mice than female WT mice particularly in the high frequencies (Fig. 2b): the thresholds in the female $Gsta4^{-/-}$ mice were approximately 40 dB higher than WT mice at 32 and 48 kHz (32 kHz, $p < 0.01$, $t = 3.485$; 48 kHz, $p < 0.001$, $t = 4.021$, two-way analyses of variance (ANOVA)). In rodents, ABRs typically consist of five positive waves; wave I represents activity from the auditory nerve, while waves II–V represent neural transmission within the central auditory system[30]. To further assess hearing sensitivity and the functional integrity of the auditory nerve, ABR latencies and amplitudes for wave I were measured with a click stimulus of 100 dB SPL in male and female WT and $Gsta4^{-/-}$ mice under cisplatin treatment (Fig. 2d, e). In cisplatin-treated male mice, there were no differences in wave I amplitudes or latencies between WT and $Gsta4^{-/-}$ mice. However, cisplatin-treated female $Gsta4^{-/-}$ mice displayed a 36% decrease in wave I amplitudes and a 3% increase in wave I latencies compared to WT mice, indicating a deficit in auditory neural processing in female $Gsta4^{-/-}$ mice. Taken together, these physiological results indicate that under cisplatin treatment, loss of $Gsta4$ results in more profound hearing loss in female mice than male mice.

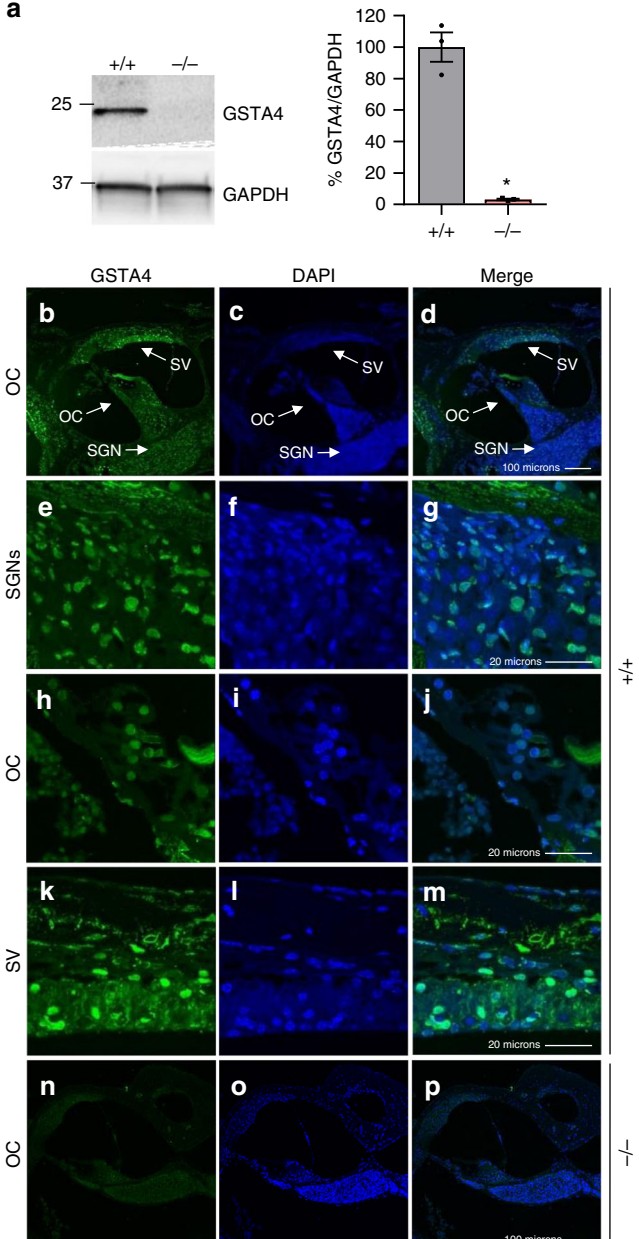

**Fig. 1** Localization of GSTA4 protein in mouse cochlea. **a** Western blot analysis of GSTA4 protein levels in the inner ear from young male $Gsta4^{+/+}$ and $Gsta4^{-/-}$ mice. The full-length blot is presented in the Source Data file. $n = 3$, *$p < 0.05$ vs. +/+ (unpaired two-tailed Student's $t$ test). Error bars represent ± s.e.m. **b–p** GSTA4 staining (green; **b**, **e**, **h**, **k**, **n**), DAPI staining (blue; **c**, **f**, **i**, **l**, **o**), and merged staining (**d**, **g**, **m**, **p**) were detected in the organ of Corti regions (**b–d**, **n–p**), SGNs (**e–g**), organ of Corti (**h–j**), and SVs (**k–m**) from 3-month-old WT (**b–m**) and $Gsta4^{-/-}$ (**n–p**) males. OC organ of Corti. Scale bar = 100 μm (**d**, **p**) or 20 μm (**g**, **j**, **m**)

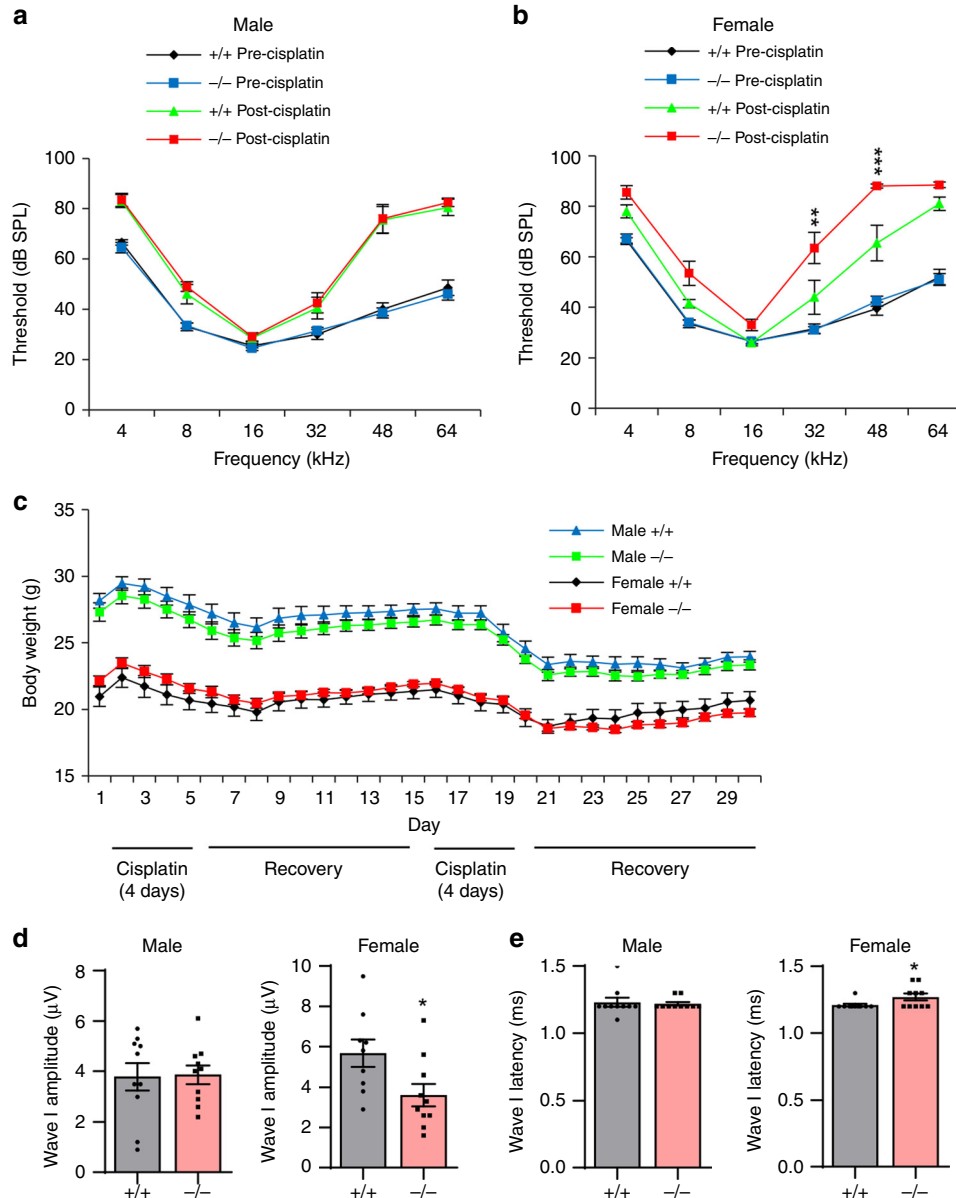

**Fig. 2** Assessment of ABR hearing thresholds, wave I amplitudes, and latencies. **a**, **b** ABR hearing thresholds were measured at 4, 8, 16, 32, 48, and 64 kHz in male (**a**) and female (**b**) $Gsta4^{+/+}$ and $Gsta4^{-/-}$ mice. $n = 10$, *$p < 0.05$ vs. +/+ post-cisplatin (two-way ANOVA). Error bars represent ± s.e.m. **c** The body weight of male and female $Gsta4^{+/+}$ and $Gsta4^{-/-}$ mice was measured during cisplatin treatment. $n = 10$. Error bars represent ± s.e.m. **d**, **e** ABR wave I amplitudes (**d**) and latencies (**e**) were measured in male and female $Gsta4^{+/+}$ and $Gsta4^{-/-}$ mice. $n = 10$, *$p < 0.05$ vs. +/+ (unpaired two-tailed Student's $t$ test). Error bars represent ± s.e.m. Source data are provided as a Source Data file

**Cisplatin causes more profound loss of cochlear cells in $Gsta4^{-/-}$ female mice.** To identify the sites of cisplatin-induced cochlear pathologies in $Gsta4^{-/-}$ mice, we counted the numbers of SGNs in the apical, middle, and basal regions of the cochlea from cisplatin-treated male and female WT and $Gsta4^{-/-}$ mice. In male mice, there were no differences in SGN densities in the apical, middle, or basal cochlear regions between cisplatin-treated WT and $Gsta4^{-/-}$ mice (Fig. 3a–g). However, cisplatin-treated female $Gsta4^{-/-}$ mice displayed a 34–41% decrease in SGN densities in the apical and basal cochlear regions compared to WT mice (apical, $p < 0.01$, $t = 4.158$; basal, $p < 0.05$, $F_{(3, 9)} = 2.678$, two-way ANOVA) (Fig. 3h–n). Because robust GSTA4 immunostaining was detected in the SV (Fig. 1k–m), we investigated whether loss of $Gsta4$ leads to reduced SV thickness in the apical, middle, and basal cochlear regions of female or male mice under cisplatin treatment. In cisplatin-treated male mice,

there were no differences in SV thickness in the apical, middle, or basal cochlear regions between WT and $Gsta4^{-/-}$ mice (Fig. 4a–g). However, in general, SV thickness was substantially lower in female $Gsta4^{-/-}$ mice versus female WT mice, particularly in the apical region under cisplatin treatment (Fig. 4h–n): cisplatin-treated female $Gsta4^{-/-}$ mice displayed a 64% decrease in SV thickness in the apical cochlear regions compared to WT mice ($p < 0.001$, $t = 4.786$, two-way ANOVA). Loss of capillaries in the SV is associated with SV atrophy[31,32]. Hence, we investigated whether $Gsta4$ deficiency leads to capillary loss in the SV in cochlear sections stained with the antibody specific to the endothelial capillary marker, endomucin. In both male and female cisplatin-treated mice, there were no differences in capillary number in the apical, middle, or basal cochlear regions between WT and $Gsta4^{-/-}$ mice (Fig. 4o–p). Lastly, to investigate whether loss of $Gsta4$ leads to HC loss in female mice under cisplatin

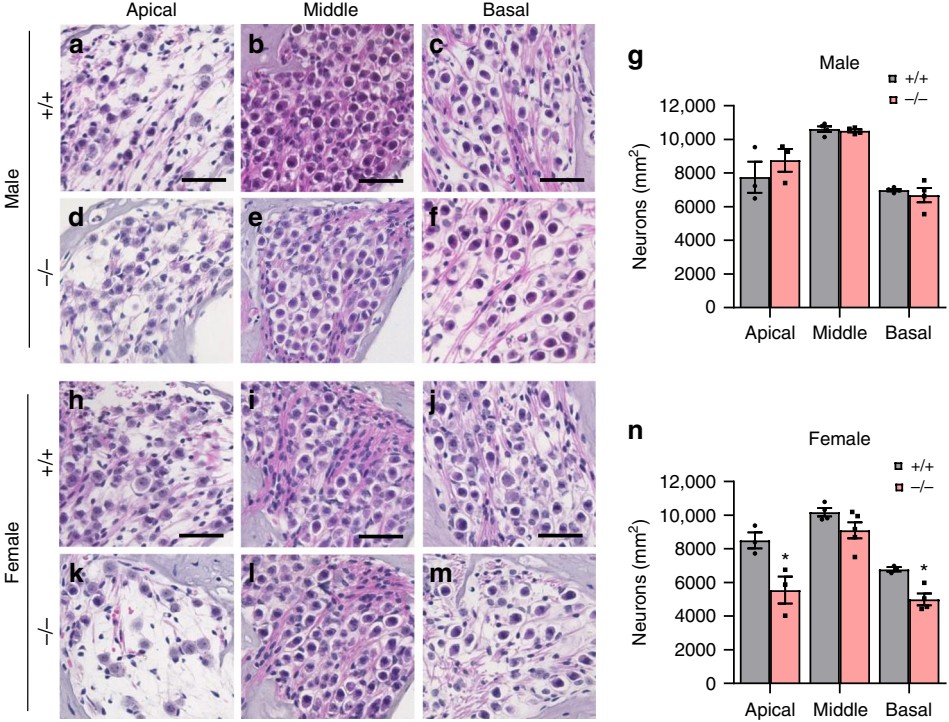

**Fig. 3** Assessment of SGN density. **a–n** SGN regions in the apical, middle, and basal regions of cochlear tissues from male (**a–f**) and female (**h–m**) $Gsta4^{+/+}$ and $Gsta4^{-/-}$ mice after cisplatin treatment. Scale bar = 20 μm. SGN densities in the apical, middle, and basal regions of cochlear tissues from male (**g**) and female (**n**) $Gsta4^{+/+}$ and $Gsta4^{-/-}$ mice were counted and quantified. The quantification shows a mean of at least three independent experiments ($n = 3$), *$p < 0.05$ vs. +/+ (two-way ANOVA). Error bars represent ± s.e.m. Source data are provided as a Source Data file

treatment, mean cochleograms were prepared from cisplatin-treated male and female WT and $Gsta4^{-/-}$ mice. In cisplatin-treated male mice, there were no differences in inner HC (IHC) loss or outer HC (OHC) loss in the apical, middle, and basal cochlear regions between WT and $Gsta4^{-/-}$ mice (Supplementary Fig. 4a–h). In female mice, however, cisplatin-treated $Gsta4^{-/-}$ mice displayed a 42% IHC loss in the basal cochlear region compared to WT mice with negligible IHC loss (90–100%, $p < 0.0001$, $t = 8.896$, two-way ANOVA) (Supplementary Fig. 5a–g). OHC loss was higher in female $Gsta4^{-/-}$ mice versus female WT mice (Supplementary Fig. 5a–f, h) in the basal regions under cisplatin treatment, but the differences were not statistically different. Together, these histological analyses indicate that under cisplatin treatment, loss of $Gsta4$ results in more pronounced cochlear pathologies in female mice compared to male mice.

**Cisplatin stimulates GSTA4 activity in the inner ears of female mice.** Previous studies have shown that GSTA4 has high-catalytic efficiency toward 4-HNE, a cytotoxic end product of lipid peroxidation[9–11]. Hence, we hypothesized that GSTA4 might reduce cisplatin ototoxicity by removing 4-HNE from cochlear cells in female mice. To test this hypothesis, we first measured GST activities using 4-HNE as a specific substrate for GSTA4 and 1-chloro-2,4-dinitorbenzene (CDNB), a general substrate for most GSTs[10], in inner ear tissues from control and cisplatin-treated male and female WT and $Gsta4^{-/-}$ mice. In male WT mice, there were no differences in GST activities toward 4-HNE between control (precisplatin treatment) and cisplatin-treatment mice (Fig. 5a). However, in female WT mice, cisplatin treatment resulted in a 47% increase of GST activities toward 4-HNE, indicating that cisplatin stimulates GSTA4 activity in female mice, but not in male mice (Fig. 5b). In both male and female $Gsta4^{-/-}$ mice, GST activities toward 4-HNE were substantially lower (~75% decrease) compared to WT mice (Fig. 5a, b), validating previous reports that GSTA4

has high catalytic efficiency toward 4-HNE[9–11]. There were no differences in GST activities toward 4-HNE between control and cisplatin-treated male or female $Gsta4^{-/-}$ mice. In measurement of GST activities toward CDNB, there were no differences in GST activities toward CDNB between control male or female WT and cisplatin-treated WT mice, and control male or female $Gsta4^{-/-}$ and cisplatin-treated $Gsta4^{-/-}$ mice (Supplementary Fig. 6a, b), validating that GSTA4 has no or little catalytic efficiency toward CDNB[10]. Next, to investigate whether GSTA4 reduces 4-HNE levels in cochlear cells under cisplatin, we counted 4-HNE-positive SGNs in cochlear sections from cisplatin-treated male and female WT and $Gsta4^{-/-}$ mice. In cisplatin-treated male mice, there were no differences in 4-HNE-positive SGN levels in the apical, middle, or basal cochlear regions between WT and $Gsta4^{-/-}$ mice (Fig. 5c). However, cisplatin-treated female $Gsta4^{-/-}$ mice displayed a 17–22% increase in 4-HNE-positive SGN levels in the apical, middle, and basal cochlear regions compared to female WT mice (apical, $p < 0.001$, $t = 5.245$; middle, $p < 0.001$, $t = 5.16$; basal, $p < 0.001$, $t = 4.777$, two-way ANOVA) (Fig. 5d), indicating that GSTA4 reduced cisplatin-induced 4-HNE in the cochlea of female mice, but not male mice.

Previous studies have reported gender differences in the gene expression of various GST isoforms, including $Gsta4$, in various tissues[33–36]. Estrogen modulates the expression of GST genes in various tissues[33,37]. Thus, we hypothesized that the constitutive $Gsta4$ gene expression in the inner ears of female mice might be higher than that of male mice due to higher levels of estrogen in females. To test this hypothesis, we performed gene-expression analysis of inner ear tissues from 5-month-old male, female (ovary-intact female), and ovariectomized female (OVX) CBA/CaJ mice using pathway-focused-PCR arrays containing 84 genes involved in Phase II detoxification. Female CBA/CaJ mice displayed higher mRNA expression of nine genes involved in Phase II detoxification, including $Gsta4$, $Gstm2$, and $Gstt1$,

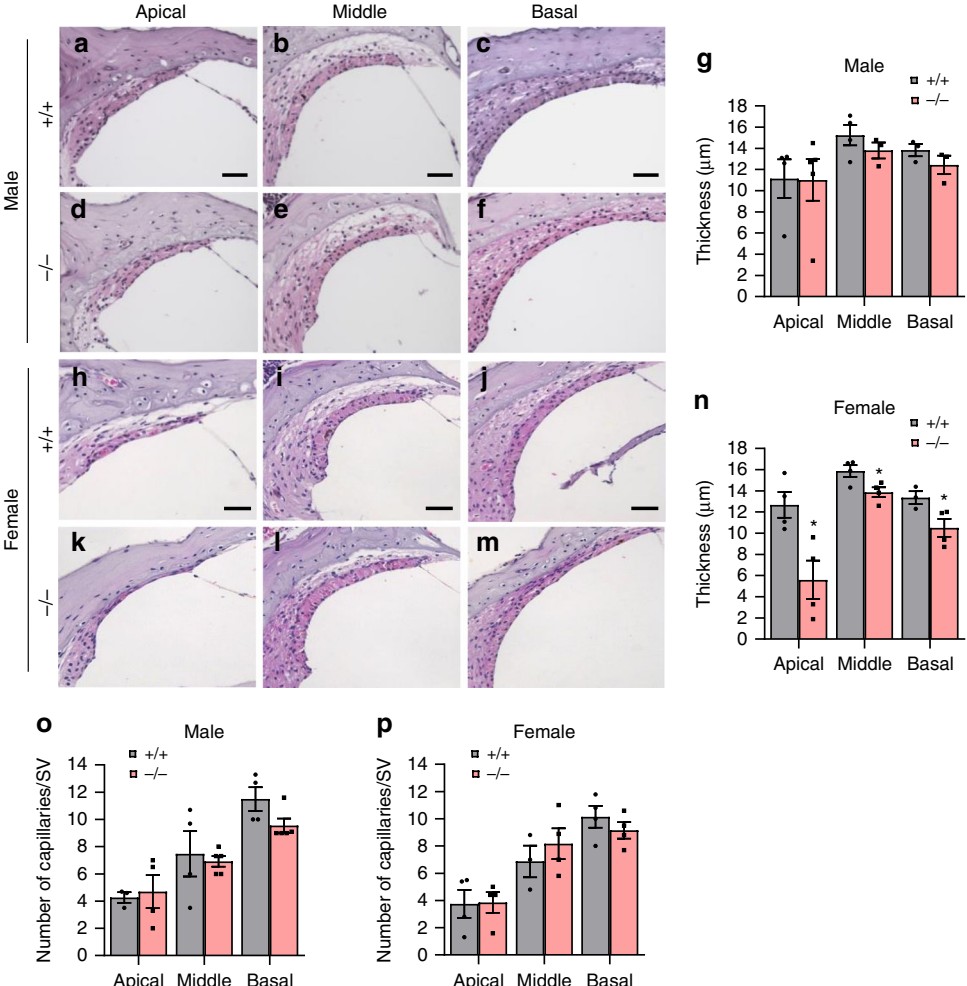

**Fig. 4** Assessment of SV thickness and capillary number. **a–n** SV regions in the apical, middle, and basal regions of cochlear tissues from male (**a–f**) and female (**h–m**) Gsta4$^{+/+}$ and Gsta4$^{-/-}$ mice after cisplatin treatment. Scale bar = 20 μm. The thickness of SV in the apical, middle, and basal region of cochlear tissues from male (**g**) and female (**n**) Gsta4$^{+/+}$ and Gsta4$^{-/-}$ was measured. The quantification shows a mean of at least three independent experiments (n = 3), *p < 0.05 vs. +/+ (two-way ANOVA). Error bars represent ± s.e.m. **o, p** Numbers of endomucin-positive capillaries in the SVs in each region (apical, middle, and basal region) of cochlear tissues were counted and quantified in male (**o**) and female (**p**) Gsta4$^{+/+}$ and Gsta4$^{-/-}$ mice after cisplatin treatment. The quantification shows mean of at least three independent experiments (n = 3). Error bars represent ± s.e.m. Source data are provided as a Source Data file

compared to males (Fig. 6a). In contrast, ovariectomy down-regulated 25 genes involved in Phase II detoxification, including *Gsta4, Gstk1, Gstm2, Gstm4, Gstp1, Gstt1, and Gstt2* (Fig. 7a), suggesting that ovarian estrogen modulates gene expression of several GST genes, including *Gsta4*. Next, we measured GSTA4 protein levels in the inner ear tissues from the same groups of CBA/CaJ mice by capillary electrophoresis-based immunoassay using a Wes system. Although there were no differences in GSTA4 protein levels between male and female CBA/CaJ mice, OVX mice showed significantly lower GSTA4 protein levels compared to female mice (Fig. 8a, b). The nuclear transcription factor E2-related factor 2 (NRF2) promotes the transcriptional induction of antioxidant genes such as the subunits of glutamate cysteine ligase, the rate-limiting enzyme in glutathione biosynthesis, and Phase II detoxification genes, including *GSTA, GSTM,* and *GSTP*. Thus, NRF2 plays a critical role in reducing xenobiotic-induced toxicity[38–40]. 4-HNE also acts as a direct activator of NRF2[41]. Moreover, earlier studies have shown that estrogen receptor α (ER1) interacts with NRF2 signaling in breast cancer cell lines[42] and ovarian cancer patients[43]. Thus, to investigate whether NRF2 interacts with GSTA4 and/or whether

the constitutive expression of *Nrf2* gene in the inner ears of female mice is higher than that of male mice, we measured mRNA expression levels of *Nrf2* in the inner ears of male, female, and OVX CBA/CaJ mice by quantitative RT-PCR. Interestingly, female CBA/CaJ mice displayed higher mRNA expression of *Nrf2* compared to males (Fig. 6b), while ovariectomy downregulated *Nrf2* in the inner ears (Fig. 7b), suggesting that ovarian estrogen also modulates gene expression of *Nrf2*, known to induce Phase II detoxification genes, including *Gsta4*.

It is well-documented that cisplatin binds to and damages nuclear DNA, leading to oxidative DNA damage[17,20]. Hence, we measured levels of 8-oxoguanine (8-oxoG), a common product of oxidative DNA damage, in inner ear tissues from cisplatin-treated male and female WT and Gsta4$^{-/-}$ mice. In both male and female mice, loss of Gsta4 resulted in a 33–51% increase in 8-oxoG levels in the inner ear (Fig. 9a). However, there were no differences in 8-oxoG levels between male and female WT mice or male and female Gsta4$^{-/-}$ mice, suggesting that GSTA4 plays a role in reducing cisplatin-induced oxidative DNA damage in the inner ears of both male and female mice. To further investigate whether ovariectomy affects cisplatin-induced oxidative DNA

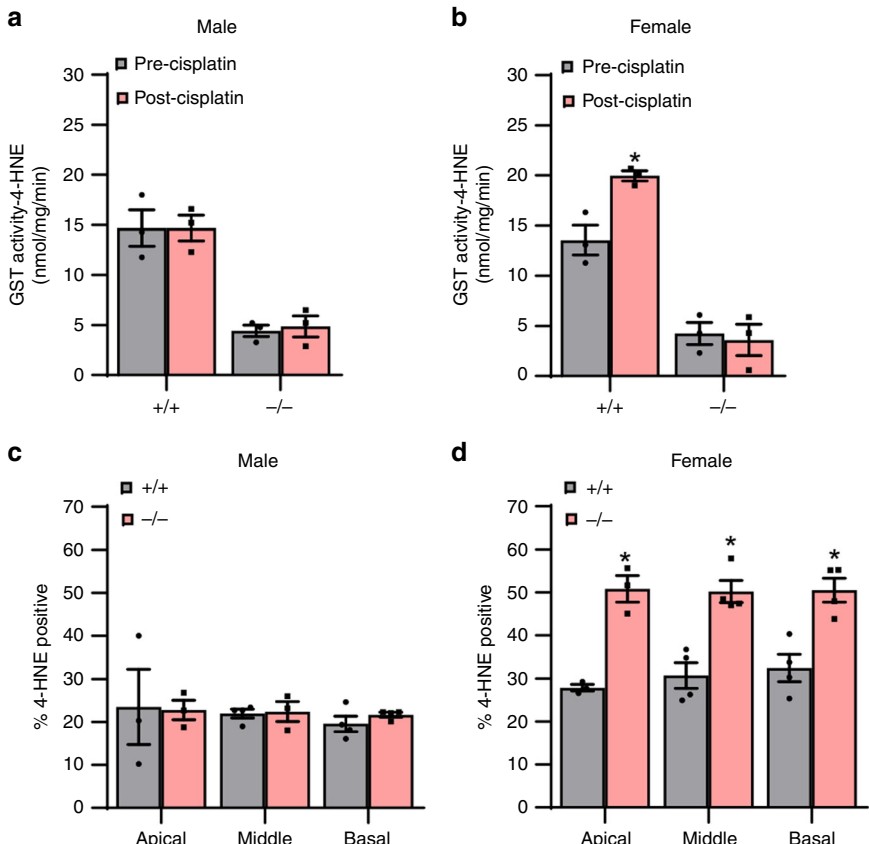

**Fig. 5** Assessment of GST activity. **a**, **b** GST activities toward 4-HNE were measured in the cytosol of inner ear tissues from male (**a**) and female (**b**) $Gsta4^{+/+}$ and $Gsta4^{-/-}$ mice prior to and after cisplatin treatment. $n = 3$, $*p < 0.05$ vs. precisplatin (two-way ANOVA). Error bars represent ± s.e.m. **c**, **d** 4-HNE-positive SGNs were counted and quantified in the apical, middle, and basal regions of cochlear tissues from male (**c**) and female (**d**) $Gsta4^{+/+}$ and $Gsta4^{-/-}$ mice after cisplatin treatment. The quantification shows a mean of at least three independent experiments ($n = 3$), $*p < 0.05$ vs. $+/+$ (two-way ANOVA). Error bars represent ± s.e.m. Source data are provided as a Source Data file

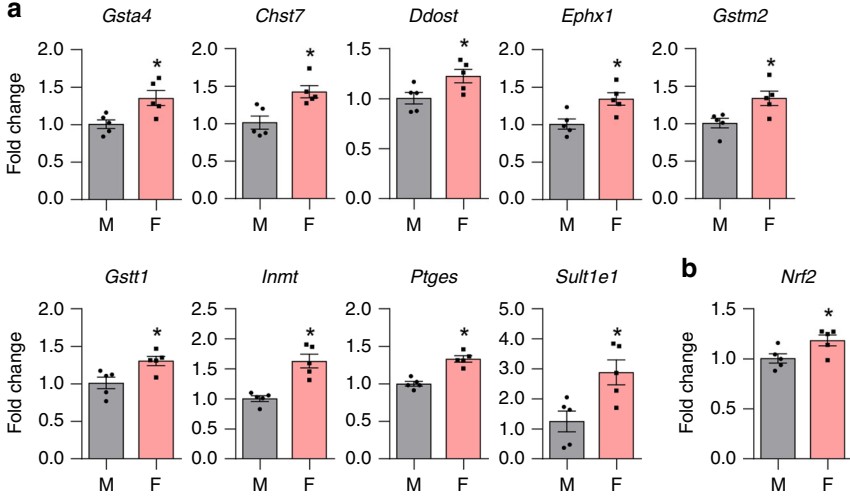

**Fig. 6** Assessment of mRNA expression levels of detoxification genes in the inner ears of CBA/CaJ mice. **a** mRNA expression levels of nine genes involved in the enzymatic processes of Phase II detoxification were analyzed by PCR arrays in the inner ears from male and female CBA/CaJ mice. $n = 5$, $*p < 0.05$ vs. M (unpaired two-tailed Student $t$ test). Error bars represent ± s.e.m. **b** mRNA expression levels of $Nrf2$ were measured by qRT-PCR in the inner ears from male and female CBA/CaJ mice. $n = 5$, $*p < 0.05$ vs. M (unpaired two-tailed Student $t$ test). Error bars represent ± s.e.m. The relative mRNA levels were normalized to levels of $B2m$. M: male, F: female. Source data are provided as a Source Data file

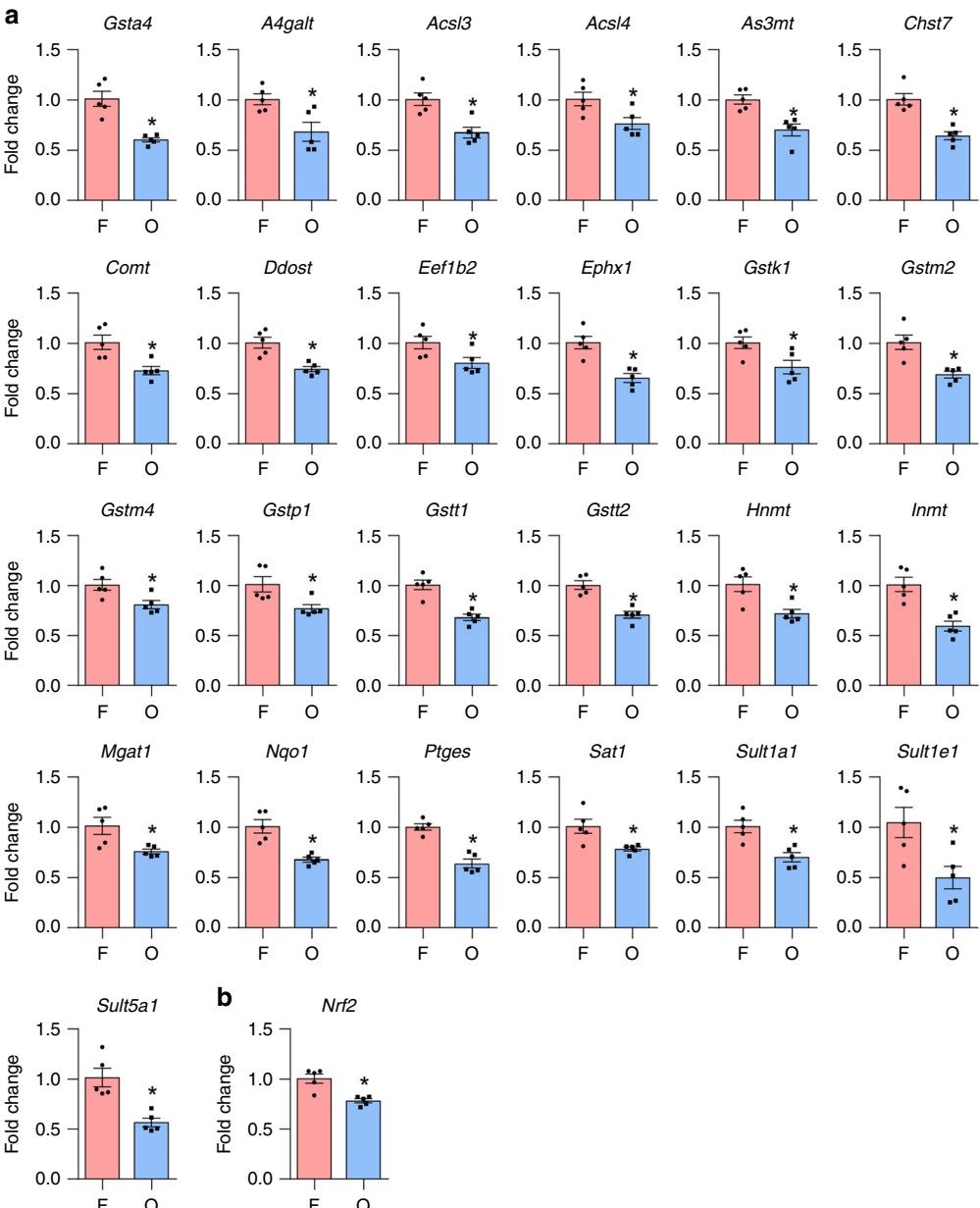

**Fig. 7** Assessment of mRNA-expression levels of detoxification genes in the inner ears of CBA/CaJ mice. **a** mRNA expression levels of 25 genes involved in the enzymatic processes of Phase II detoxification were analyzed by PCR arrays in the inner ears from female and OVX CBA/CaJ mice. $n = 5$, *$p < 0.05$ vs. F (unpaired two-tailed Student $t$ test). Error bars represent ± s.e.m. **b** mRNA expression levels of Nrf2 were measured by qRT-PCR in the inner ears from female and OVX CBA/CaJ mice. $n = 5$, *$p < 0.05$ vs. F (unpaired two-tailed Student $t$ test). Error bars represent ± s.e.m. The relative mRNA levels were normalized to levels of B2m. F: female, O: ovariectomized female. Source data are provided as a Source Data file

damage in female Gsta4$^{-/-}$ mice, we measured 8-oxoG levels in the inner ears from cisplatin-treated male, female, and OVX Gsta4$^{-/-}$ mice. There were no differences in 8-oxoG levels between male and female Gsta4$^{-/-}$ mice or female and OVX Gsta4$^{-/-}$ mice (Fig. 9b, c). We also measured protein carbonyl levels, a marker of oxidative protein damage, in the inner ears from cisplatin-treated female, male and OVX Gsta4$^{-/-}$ mice. There were also no differences in levels of protein carbonyl between male and female Gsta4$^{-/-}$ mice or female and OVX Gsta4$^{-/-}$ mice (Supplementary Fig. 7a-b). Glutathione (γ-glutamyl–cysteinyl–glycine) plays an important role in anti-oxidant defense[44] and also serves as a substrate for GSTs in detoxifying toxins[3]. To investigate the role of GSTA4 in reducing oxidative damage induced by cisplatin, we measured the levels of

oxidized glutathione (GSSG), reduced glutathione (GSH), and total glutathione in the cytosol of the inner ear tissues from cisplatin-treated male and female WT and Gsta4$^{-/-}$ mice. In both male and female mice, there were no differences in GSH or GSSG levels or GSH/GSSG ratios between WT and Gsta4$^{-/-}$ mice (Supplementary Fig. 8a, b), indicating that Gsta4 deficiency does not affect the glutathione redox status in the inner ear of both male and female mice.

## Discussion
Our results indicate that cisplatin stimulates GSTA4 activity in the inner ear of female WT, but not male WT mice. Under cis-platin treatment, loss of Gsta4 results in more profound loss of

cochlear cells and more profound hearing loss in female mice compared to male mice. The question then becomes how does cisplatin stimulate GSTA4 activity in the inner ear of females, but not males? Numerous studies have shown that constitutive gene expression of GST isoforms is tissue specific in humans and rodents[35,45,46]. For example, *Gsta3* mRNA was most highly expressed in the liver, while lower levels of *Gsta3* mRNA were observed in the kidney, brain, and heart in mice[35]. mRNA expression of *Gsta4* was highest in the stomach, while lower levels of *Gsta4* mRNA were observed in the liver and kidney. Moreover, the expression of various GST isoforms is altered in response to cancer drugs and during carcinogenesis[6,24], suggesting that the expression of GSTs can influence the efficacy of anticancer drugs[3,45]. Several studies have reported gender differences in the gene expression of various GST isoforms[33–36]: female mice

showed a 12-fold higher mRNA expression of *Gsta4* in the kidney and a 2.5-fold higher mRNA expression of *Gsta4* in the heart compared to males[35]. In the kidney, mRNA expression levels of *Gsta1/2*, *Gsta3*, *Gstk1*, *Gstm2*, and *Gsto1* were also higher in females than males, and the same trend was observed for heart. These reports suggest that the gene expression of GSTs, including *Gsta4*, is regulated or influenced by estrogen. This idea is supported by previous reports showing that estrogen interacts with GSTs in different tissues[33,37]. In the current study, female CBA/CaJ mice, a model of sex differences in hearing and longevity, displayed higher mRNA expression of *Gsta4*, compared to males, while ovariectomy downregulated *Gsta4* in the inner ears, indicating that the constitutive expression of *Gsta4* in the inner ears of female mice is higher than that of male mice. These results also suggest that ovarian estrogen modulates gene expression of *Gsta4*. Thus, we speculate that cisplatin or increased levels of 4-HNE induced by cisplatin stimulates GSTA4 activity in the inner ear of females, but not males, likely because the constitutive expression of inner ear *Gsta4* mRNA is significantly higher in females than males and/or females have higher levels of estrogen. The other important question is how does *Gsta4* deficiency lead to more pronounced cisplatin ototoxicity in females compared to males? Previous studies have shown that GSTA1, GSTA2, GSTA4, GSTM1, and GSTT1 also possess antioxidant activity toward lipid hydroperoxides, including 4-HNE[1,3,6,7,10]. Therefore, we speculate that a complete loss of *Gsta4* leads to more pronounced cisplatin ototoxicity in females compared to males likely because: (1) females use GSTA4 as a primary detoxification enzyme to remove 4-HNE due to the higher abundance of *Gsta4*, (2) GSTA4 has higher catalytic efficiency toward 4-HNE than other GSTs[10,11], and/or (3) males do not use GSTA4 as a primary detoxification enzyme to remove 4-HNE due to the lower abundance of *Gsta4*. Lastly, our oxidative DNA damage assay results show that loss of *Gsta4* results in increased oxidative DNA damage in the inner ears of both males and female mice, suggesting a role of GSTA4 in reducing cisplatin-induced oxidative DNA damage in both sexes. We speculate that GSTA4 can mediate a reduction of cisplatin-induced oxidative DNA damage because 4-HNE is known to covalently modify DNA[9,12] and GSTA4 has high catalytic efficiency toward 4-HNE[10,11]. However, the mechanism by which *Gsta4* deficiency results in increased oxidative DNA damage in both sexes remains unclear.

It is well-documented that women live longer than men in every country in the world[47]. A similar pattern of sex differences in longevity is found in many other animals. Numerous studies have also reported gender differences in human auditory

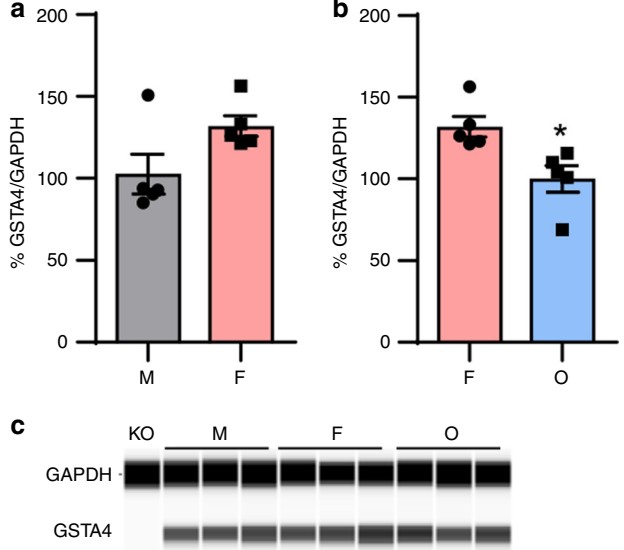

**Fig. 8** Assessment of GSTA4 protein levels in the inner ears of CBA/CaJ mice. **a**, **c** GSTA4 protein levels were measured in the inner ear tissues from male and female CBA/CaJ mice. $n = 5$. Error bars represent ± s.e.m. **b**, **c** GSTA4 protein levels were measured in the inner ear tissues from female and OVX CBA/CaJ mice. $n = 5$, *$p < 0.05$ vs. F (unpaired two-tailed Student *t* test). Error bars represent ± s.e.m. The relative protein levels were normalized to levels of GAPDH. The full-length blot is presented in the Source Data file. M: male, F: female, O: ovariectomized female. Source data are provided as a Source Data file

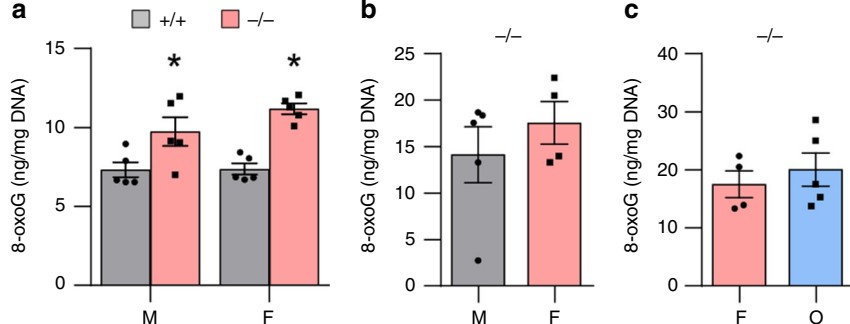

**Fig. 9** Assessment of oxidative DNA damage in the inner ears of *Gsta4*[+/+] and *Gsta4*[−/−] mice. **a** Levels of 8-oxoguanine as an oxidative DNA damage marker were measured in the inner ear tissues from cisplatin-treated male and female *Gsta4*[+/+] and *Gsta4*[−/−] mice. $n = 5$, *$p < 0.05$ vs. +/+ (unpaired two-tailed Student *t* test). Error bars represent ± s.e.m. **b** Levels of 8-oxoguanine were measured in the inner ears from cisplatin-treated male and female *Gsta4*[−/−] mice. The quantification shows a mean of at least four independent experiments ($n = 4$). Error bars represent ± s.e.m. **c** Levels of 8-oxoguanine were measured in the inner ears from cisplatin-treated female and OVX *Gsta4*[−/−] mice The quantification shows a mean of at least four independent experiments ($n = 4$). Error bars represent ± s.e.m. M: male, F: female, O: ovariectomized female. Source data are provided as a Source Data file

perception. In general, the results of these studies show that women lose hearing more slowly than men and that women of virtually all ages demonstrate better hearing than age-matched men[48–53]. Considerable evidence also suggests that auditory function is diminished following menopause, whereas estrogen replacement therapy lowers hearing thresholds, shortens ABR latencies, and increases ABR amplitudes in postmenopausal women[51,54,55]. Importantly, long-term exposure of human breast cancer cells to estrogen increases their sensitivity to cisplatin[56]. Given that estrogen has antioxidant and neuroprotective actions[57–60], higher levels of estrogen might underlie a female survival advantage. Previous studies have shown that knockdown of either gst-5 or gst-10, which encode GST proteins that detoxify 4-HNE, reduces HNE-conjugating activity and lifespan[13], while overexpression of gst-10 increases HNE-conjugating activity and lifespan in C. elegans[14]. McElwee et al.[61] conducted a cross-species analysis to compare gene expression changes in long-lived C. elegans (daf-2), Drosophilla melanogaster (chico1/+), Ames dwarf mice (Prop1df/df), and Little dwarf mice (Ghrhrlit/lit). The authors found that GST and other cellular detoxification gene categories were significantly up-regulated in long-lived members of the four species, suggesting that those long-lived animals have enhanced GST detoxification. In the current study, our PCR array results suggest that female CBA/CaJ mice have enhanced GST detoxification, whereas GST detoxification function is diminished following ovariectomy. Our quantitative RT-PCR results also show that female CBA/CaJ mice have elevated levels of Nrf2 that promotes the transcriptional induction of a number of Phase II detoxification genes, including Gsta, Gstm, and Gstp. In C. elegans, the skn-1 gene encodes a transcription factor that resembles mammalian Nrf2 transcription factors and activates a detoxification response[38,62]. Importantly, loss of skn-1 shortens lifespan, while skn-1 over-expression increases lifespan in C. elegans, suggesting that Nrf2-mediated detoxification may play a role in longevity. Thus, we speculate that NRF2-mediated GSTA4 detoxification may play a role in estrogen-mediated neuroprotection and sex differences in hearing and longevity.

GSTA4 is a member of the α-class GSTs and has high-catalytic efficiency in conjugating 4-HNE with GSH[10,11]. In agreement with these reports, several structural studies of GSTA4 have confirmed its high substrate specificity toward 4-hydroxyalkenals, including 4-HNE and structural differences contributing to the distinct substrate specificities of GSTAs[9]. For example, Tyr212 is proposed to be a key residue in the interaction with the aldehyde of 4-hydroxyalkenals. However, the corresponding residue is not conserved in other members of GSTAs. In contrast, there is no report that GSTA4 can catalyze GSH conjugation of cisplatin. Among other GST isoforms, several studies have shown that GSTP could detoxify cisplatin. For example, the expression of GSTP is associated with cisplatin sensitivity in nephrotoxicity and cancer cells[63,64]. Although the catalytic mechanism of GSTPs remains unclear, a cisplatin-conjugated structure of human GSTP1 is available from Protein Data Bank (PDB code, 5DJL). Based on the structure, two cysteine residues near the GSH binding site are required to hold the platinum atom from cisplatin. However, those cysteine residues are not observed in the structure of GSTA4, suggesting that it is not possible for GSTA4 to function in detoxification of cisplatin in the same manner as GSTP. In the current study, we demonstrated that cisplatin treatment increased GSTA4 activity toward 4-HNE in the cochlea of WT female mice. In female Gsta4−/− mice, cisplatin treatment also resulted in increased levels of 4-HNE in the SGNs. In agreement with our data, cisplatin administration increased 4-HNE levels in the kidney of rats[21]. Therefore, our results and previous reports indicate that GSTA4 reduces cisplatin ototoxicity

by removing 4-HNE, but not by directly removing cisplatin in mouse cochlea.

A large body of evidence shows that degeneration of SV is one of the major causes of cisplatin-induced ototoxicity. In guinea pigs, cisplatin administration resulted in degeneration of stria vascularis which was associated with a decline in the endocochlear potential and degeneration of marginal cells[65]. Another guinea pig study showed a three to fivefold accumulation of DNA adduct (guanine–guanine intrastrand cross-link) exclusively in the marginal cells of the stria vascularis after treatment with cisplatin[66]. In humans, histopathological examination of temporal bone specimens from patients who received cisplatin revealed a significant loss of IHCs and OHCs, atrophy of the stria vascularis, and decreased SGNs in the basal turn of the cochlea[67]. In cisplatin-treated rats, transmission electron microscopy examination revealed strial edema, rupture of marginal cells, and depletion of the cytoplasmic organelles in the cochlea[68]. A recent study has shown that while cisplatin is eliminated in most organs over days to weeks, cisplatin remained in the cochlea for months to years in both mice and humans[23]. Interestingly, cisplatin levels were higher in the SV compared to the OC or SG region. In agreement with these reports, we observed very intense GSTA4 immunostaining in the SV (Fig. 1b-m) and moderate to severe SV atrophy in female Gsta4−/− mice after cisplatin treatment (Fig. 4h–n). These results suggest that the SV is likely a major site for GSTA4 detoxification activity in female mice. In summary, our findings identify GSTA4 as an essential player in reducing cisplatin ototoxicity in females.

## Methods

**Experimental animals.** Male and female Gsta4+/− mice were obtained from the Mutant Mouse Regional Resource Centers (MMRRC)[69]. CBA/CaJ mice were purchased from the Jackson Laboratory (Bar Harbor, ME). Gsta4+/− mice were backcrossed onto the CBA/CaJ mice for six generations and the N6 heterozygous CBA/CaJ-Gsta4+/− mice were subsequently intercrossed to generate CBA/CaJ-Gsta4+/+, CBA/CaJ-Gsta4+/−, and CBA/CaJ- Gsta4−/− littermates. Both male and female CBA/CaJ-Gsta4+/+ and CBA/CaJ-Gsta4−/− were used in the current study. All animal experiments were conducted under protocols approved by the University of Florida Institutional Animal Care and Use Committee. We have complied with all relevant ethical regulations.

**Genotyping.** CBA/CaJ-Gsta4+/− males were mated with CBA/CaJ-Gsta4+/− females and their offspring were genotyped by PCR with the DNA extracted from the tails of these mice. Primer sequences and cycling conditions for PCR were as follows:

Gsta4 Forward 5′-GGAAACTGGCTGTGCGCTAC-3′;
Gsta4 Reverse: 5′-TCAGCTCCGGCTCTCTTCTG-3′;
Gsta4 KO Reverse 5′-ATAAACCCTCTTGCAGTTGCATC-3′;

94 °C for 2 min; 35 cycles of 94 °C for 30 s, 58 °C for 20 s, and 72 °C for 30 s with an extension at 72 °C for 5 min. PCR products were separated on 1.5% agarose gel and the expected band size for the WT and knockout allele were 135 and 140 bps, respectively. The full-length gel is presented in the Source Data file.

The Gsta4+/− mice provided by the MMRRC were originally developed on the 129S5/SvEvBrd;C57BL/6J background[69]. Because C57BL/J mice are homozygous for the age-related hearing loss (AHL)-susceptibility allele (Cdh23753A)[25], the Gsta4+/− mice were backcrossed for six generations onto the CBA/CaJ mouse strain that does not carry the early-onset AHL-susceptibility allele (Cdh23753A)[26]. To confirm that N6 CBA/CaJ-Gsta4+/+ and CBA/CaJ-Gsta4−/− mice have the same WT Cdh23753G/753G genotype, we isolated DNA from the tails of these mice, amplified by PCR, and then sequenced the region of DNA containing the 753rd nucleotide in the Cdh23 gene. Primer sequences and cycling conditions for PCR were as follows:

Cdh23 Forward 5′- GATCAAGACAAGACCAGACCTCTGTC-3′;
Cdh23 Reverse 5′-GAGCTACCAGGAACAGCTTGGGCCTG-3′;

95 °C for 2 min; 35 cycles of 95 °C for 30 s, 60 °C for 1 min, 72 °C for 1 min; with an extension at 72 °C for 5 min. The expected band size of the PCR product was 360 bps. The Cdh23 gene in three CBA/CaJ-Gsta4+/+ and three CBA/CaJ Gsta4−/− mice was sequenced. All the mice examined had the same WT Cdh23753G/753G genotype.

**Isolation of cytosol.** Labyrinth tissues including the bony shell, cochlear lateral wall, cochlear basilar membrane, cochlear modiolus, utricle, saccule, and three semicircular canals were homogenized using a tissue grinder (Wheaton Dounce

Tissue Grinder, Fisher Scientific) containing 1 ml of Tris buffer (10 mM Tris, 1 mM EDTA, 320 mM sucrose, pH 7.4) on ice and then centrifuged at $15,000 \times g$ for 10 min at 4 °C. The supernatant (cytosolic fraction) was used for measurement of GST activity and western blotting. For the GST activity toward 4-HNE, tissues were homogenized in 1X CPE buffer (20 mM MOPS, 0.1 mM EDTA, 62 mM sucrose, pH 7.2) on ice and then centrifuged at $15,000 \times g$ for 10 min at 4 °C.

**Western blotting**. Thirty-five micrograms of total protein were fractionated by 4–20% SDS-PAGE and transferred to nitrocellulose membranes (Bio-Rad). Membranes were incubated with the primary antibody followed by the horseradish peroxidase-linked secondary antibody. A chemiluminescent detection reagent (ECL Prime, GE Healthcare Life Sciences) was used to visualize proteins. The band intensity was quantified using the ImageJ software (National Institutes of Health) and the levels of each protein were normalized by loading controls. Primary antibodies used were as follows: Gsta4 (rabbit polyclonal, used at 1:1000 dilution, Sigma, SAB1401164) and GAPDH (rabbit polyclonal, used at 1:50,000 dilution, Sigma, G9545). Secondary antibody was goat anti-rabbit IgG (H + L) (1:5000 dilution, GE Healthcare Life Sciences, NA934). The full-length blot is presented in the Source Data file.

**ABR hearing test**. ABR thresholds were measured with tone bursts at 4, 8, 16, 32, 48, and 64 kHz using an ABR recording system (Tucker-Davis Technologies). Mice were anesthetized with ketamine (100 mg/kg) and xylazine (10 mg/kg) by intraperitoneal injection. Subdermal needle electrodes were placed at the vertex (active), ipsilateral ear (reference), and contralateral ear (ground). At each frequency, the sound level was reduced in 5–10 dB increments from 90 to 10 dB sound pressure level (SPL). A hearing threshold was defined as the lowest level that produced a noticeable ABR. ABR latencies and amplitudes for wave I were also measured with a click stimulus of 100 dB SPL. Wave I latency was determined by measuring the amount of time elapsed from the onset of the stimulus to the onset of the first ABR wave as previously described. Wave I amplitude was determined by measuring the voltage difference between the highest value (peak) and the lowest value (trough) for the first ABR wave. We used 10 male and 9–10 female mice per group for ABR threshold, wave I latency, and wave I amplitude assessments.

**Body weight**. The body weight of the male and female $Gsta4^{+/+}$ and $Gsta4^{-/-}$ mice was measured daily during the $3 \times 4$ cisplatin protocol. We used ten male and ten female mice per group for body weight measurements.

**Cisplatin treatment**. ABR tests were performed in 2–3 months old mice 1 week before the first cisplatin injection. Following the ABR tests, mice received 2–3 ml saline (Teknova, Fishersci) 1 day before the first cisplatin injection. Cisplatin (100 mg/ml; APP pharmaceuticals) was administered i.p. at 3 mg/kg each day for 4 days using a modified cisplatin protocol described by Lisa Cunningham's group[19,23]. Following this 4-day cisplatin injection period, mice recovered for 10 days. Mice received 2–3 ml saline daily during the 4-day cisplatin injection period and daily during the 10-day recovery period. Mice were also provided with a high calorie liquid diet (STAT, PRN Pharmacal) and bacon softies (BioServ) daily. This 14-day protocol was repeated once more for a total of two cycles of cisplatin administration. Mice had an additional 7 days of recovery following the final cisplatin injection before the postcisplatin treatment ABR tests. To promote exercise and facilitate the study of healthy animals, mice were housed in cages with mouse Igloos (hut) with attached wheel (BioServ) throughout the cisplatin administration protocol.

**Cochlear pathology**. After the postcisplatin treatment ABR hearing tests, the mice were sacrificed by cervical dislocation. Temporal bones were excised from the head and divided into cochlear and vestibular parts. For cochleograms, the excised cochlea was immersed in 10% formalin for 1 day. To make paraffin-embedded cochlear sections, the excised cochlea was immersed in 4% paraformaldehyde in PBS for 1 day, decalcified in 10% EDTA, pH 7.4, in PBS for 5–7 days, and embedded in paraffin. The paraffin-embedded cochlear specimens were sliced into 5 µm sections, mounted on silane-coated slides, stained with hematoxylin and eosin (H&E), and observed under a light microscope (Leica Microsystems). H&E-stained cochlear sections were used for SGN counting and SV thickness measurements. Unstained cochlear sections were used to count 4-HNE-positive cells.

The numbers of IHCs, first-row OHC1, second-row OHC2, and third-row OHC3 were counted over 0.24 mm intervals from the apex to the base of the cochlea using a microscope at 400× magnification. The counting results were then entered into a custom computer program designed to compute a cochleogram that shows the percentage of missing IHC and OHC as a function of percentage distance from the apex of the cochlea. The frequency-place map for mouse cochlea was shown on the abscissa in Fig. 8[70]. We used three to four mice per group for cochleograms.

Spiral ganglion neurons (SGNs) were counted in the apical, middle, and basal regions of the H&E-stained cochlear sections using a 40× objective. Type I and type II neurons were not differentiated, and cells were identified by the presence of a nucleus. The corresponding area of the Rosenthal's canal was measured in digital photomicrographs of each canal profile. The perimeter of the canal was traced with

a cursor using the ImageJ software (National Institutes of Health). The software then calculated the area within the outline. SGN density was measured as the number of SGNs per $mm^2$. Three to nine sections of the apical, middle, and basal turns were evaluated in one cochlea per mouse. We used three to four mice per group for SGN counting.

Stria vascularis (SV) thickness was measured in the apical, middle, and basal regions of the H&E-stained cochlear sections using a 40× objective. In the ImageJ software, the measurement was made by using a cursor to draw a line from the margin of the stria to the junction of the basal cells within the spiral ligament halfway between the attachment of Reissner's membrane and the spiral prominence. Three to nine sections of the apical, middle, and basal turns were evaluated in one cochlea per mouse. We used three to four mice per group for SV thickness measurements.

To visualize capillaries in the stria vascularis, endomucin, a marker of endothelial cells was used to stain paraffin-embedded cochlear sections. Cochlear sections were rehydrated and then incubated with rat anti-mouse endomucin antibodies (dilution 1:250; Santa Cruz Biotechnology, sc-65495). The secondary antibody (Rat on mouse AP polymer; Biocare Medical, RT518) was detected with Vulcan Fast Red Chromogen Kit 2 (Biocare Medical). Sections were counterstained with hematoxylin QS (Vector Laboratories), and endomucin-positive capillaries in the stria vascularis in the apical, middle, and basal regions of the cochlear sections were counted using a 40× objective[34].

**Immunohistochemistry**. For confocal-based immunostaining of GSTA4, cochlear sections were rehydrated before the antigen retrieval process (0.01 M Sodium Citrate pH 6.0 for 30 min at 60 °C). Sections were then incubated in diluted primary antibody (Gsta4 rabbit polyclonal, used at 1:40 dilution, MyBioSource, MBS2005646) overnight at 4 °C. The following day, the slides were washed extensively and appropriate fluorescently labeled secondary antibodies (Alexa Fluor 488 Goat Anti-Rabbit IgG (H + L), used at 1:500 dilution, Jackson ImmunoResearch Laboratories, 111-545-045) were applied for 2 h at 37 °C. The slides were then washed thoroughly and treated with DAPI (4′,6-diamidino-2-phenylindole, Thermo Fisher Scientific) for DNA visualization. Cover slips were mounted with 60% glycerol in TBS containing p-phenylenediamine (to inhibit fluorescence quench). Digital images were gathered with a Leica SP5 laser scanning confocal microscope. Figures were assembled using CorelDRAW 12 software.

**GST activity**. GST activity toward 1-chloro-2,4-dinitorbenzene (CDNB) was measured using the GST Assay Kit (Sigma-Aldrich) according to the manufacturer's instructions. In brief, 20 µl of cytosolic lysate was added to a well in the 96-well plate and then 176.4 µl of mixture containing 176.4 µl of Dulbecco's phosphate buffered saline (DPBS), 1.8 µl of 200 mM L-Glutathione reduced, and 1.8 µl of 100 mM CDNB was added to the well. The absorbance was read at 340 nm every minute for 6 min in a spectrometer (Bio-Tek) to calculate the activity. For GST activity toward 4-HNE, 10 µl of cytosolic lysate was added to a well in the 96-well plate and then 190 µl of mixture containing 5 mM reduced L-glutathione and 2 mM 4-HNE in 100 mM potassium phosphate buffer (pH 6.5). The absorbance was read at 225 nm every 30 s for 10 min in a spectrometer (Bio-Tek) to calculate activity. The GST activity was normalized to total protein. All samples were run in duplicate.

**Measurement of 4-HNE**. For immunostaining of 4-HNE-modified proteins, cochlear sections were rehydrated and incubated in Tris buffer (10 mM Tris, 1 mM EDTA, 0.05% Tween20, pH 9.0) at 90 °C for 10 min for antigen retrieval. Sections were incubated in hydrogen peroxide solution (3% $H_2O_2$, 10% MeOH in phosphate-buffered saline (PBS)) for 5 min for inactivation of endogenous peroxidase and incubated in PBS-T (0.3% TritonX-100 in PBS) for 10 min for permeabilization. Tissues were blocked in 5% normal goat serum for 30 min and incubated in primary antibody (4-HNE rabbit polyclonal, Abcam, ab46545) at 4 °C overnight. 4-HNE immunostaining was visualized with biotinylated secondary antibody, followed by avidin–biotin–peroxidase (Vectastain Elite ABC Kit, Vector Labs, PK-6100), and 3,3'-diaminobenzidine reaction. Sections were counterstained with haematoxylin and mounted in permount. 4-HNE-positive SGNs were counted in the apical, middle, and basal regions of the cochlear sections using a 40× objective and ImageJ software. The percentage of 4-HNE-positive SGNs was measured as the number of 4-HNE-positive SGNs out of the total number of SGNs. Two to four sections of the apical, middle, and basal turns were evaluated in one cochlea per mouse. We used three to four mice per group for 4-HNE-positive cell counting.

**Ovariectomy**. Female CBA/CaJ and $Gsta4^{-/-}$ mice were ovariectomized under isoflurane (Piramal Healthcare) in oxygen using a VetEquip anesthesia system. Bilateral incisions were made to expose the ovaries, which were cleared from the fat tissue and dissected out. Subsequent to the closure of the incisions, buprenorphine (0.05 mg/kg), and saline were given by subcutaneous injection.

**PCR arrays**. Total RNA was isolated from inner ear tissues obtained from male, female, and ovariectomized female CBA/CaJ mice at 5 months of age using the RNeasy Lipid Tissue Mini Kit (Qiagen, 74804). The cDNA was constructed from

500 ng of total RNA using the RT2 First Strand Kit (Qiagen, 330404). The mRNA harvested from these groups was evaluated with the Drug Metabolism: Phase II Enzymes RT$^2$ Profiler PCR Array containing 84 genes involved in the enzymatic processes of Phase II detoxification (Qiagen, PAMM-069ZD). Using the RT$^2$ SYBR Green qPCR Mastermix (Qiagen, 330502), real-time PCR was performed with a hot start to activate the DNA polymerase at 95 °C for 10 min, followed by 40 cycles of amplification (95 °C for 15 s, 60 °C for 1 min) using the CFX96 Touch Real-Time PCR Detection System (Bio-Rad). The cycle threshold (Ct) values were obtained and $2^{(-ddCt)}$ was calculated by normalizing with the housekeeping gene B2m (beta-2-microglobulin) for each gene across the PCR arrays to determine fold change. Fold change p values < 0.05 were considered to be statistically significant. The p-values were calculated based on a Student's t test of the $2^{(-ddCt)}$ values for each gene in the male group versus the same gene in the female group or for each gene in the female group versus the same gene in the ovariectomized female group. We used five mice per group for PCR array analysis.

**Quantitative RT-PCR**. cDNA was synthesized from the total RNA with the SuperScript III First-Strand Synthesis System (Thermo Fisher Scientific, 18080051) according to the manufacturer's instructions. Quantitative PCR was performed to measure mRNA levels of Nrf2/Nfe2l2 (nuclear factor, erythroid derived 2, like 2) in the inner ears of CBA/CaJ mice at 5 months of age. Relative gene expression was normalized to B2m. Reactions were performed using the TaqMan Fast Advanced Master Mix (Thermo Fisher Scientific, 4444556) on the CFX96 Touch Real-Time PCR Detection System (Bio-Rad) according to the manufacturer's instructions. The following primers and probes were used: Nfe2l2 (Thermo Fisher Scientific, TaqMan Gene Expression Assay, Mm00477784_m1) and B2m (Thermo Fisher Scientific, TaqMan Gene Expression Assay, Mm00437762_m1). We used 5 mice per group for assessment of Nrf2 expression.

**Measurement of GSTA4 protein levels**. For the capillary electrophoresis-based immunoassay, inner ear tissues were homogenized using a tissue grinder (Wheaton Dounce Tissue Grinder, Fisher Scientific) containing 1 ml of 1× Cell Lysis Buffer (20 mM Tris-HCl, pH 7.5, 150 mM NaCl, 1 mM Na2EDTA, 1 mM EGTA, 1% Triton, 2.5 mM sodium pyrophosphate, 1 mM β-glycerophosphate, 1 mM Na$_3$VO$_4$, 1 μg/ml leupeptin, Cell Signaling Technology) on ice and then centrifuged at 15,000 × g for 10 min at 4 °C. Capillary electrophoresis-based immunoassay was performed on a Wes system (ProteinSimple; San Jose, CA) according to the manufacturer's instructions using a 12–230 kDa Separation Module (ProteinSimple SM-W004 or SM-W002) and Anti-Rabbit Detection Module (ProteinSimple DM-001). Briefly, inner ear whole cell lysate samples were diluted to a concentration of 0.6 μg/μl in sample buffer (100 times diluted "10× Sample Buffer 2") and Fluorescent Master Mix from the Separation Module. Samples were heated at 95 °C for 5 min. The samples, blocking reagent (antibody diluent 2; ProteinSimple 042-203), primary antibodies (in antibody diluent 2), horseradish peroxidase (HRP)-conjugated secondary antibodies and chemiluminescent substrate were pipetted into the Separation Module plate. The primary (anti-GSTA4; 1:600 dilution; Sigma Aldrich ABS1652) and loading control (anti-GAPDH,1:20,000 dilution, Sigma G9545) antibodies were loaded in the same well as a duplex immunoassay. Instrument settings used were as follows: stacking and separation at 375 V for 25 min; blocking reagent for 10 min, primary and secondary antibody both for 30 min; luminol/peroxide chemiluminescence detection (exposures of 1,2, 4, 8, 16, 32, 64, 128, and 512 s). The high-dynamic range resulting electropherograms calculated by the software were inspected to determine whether automatic peak detection and internal standards required manual correction. The average peak signal-to-noise (S/N) ratio for GSTA4 and GAPDH calculated by the software was 194.1 ± 35.2 and 866.5 ± 180.4, respectively. The average peak height to baseline calculated manually for GSTA4 and GAPDH was 5.4 ± 0.8 and 19.2 ± 2.8, respectively. Hence, the GSTA4 and GAPDH detection peaks were distinguishable from background, and the peaks did not overlap. The full-length blot is presented in the Source Data file.

**Measurement of oxidative damage markers**. Levels of the oxidative DNA damage marker, 8-oxoguanine (8-oxoG), were measured using the Oxiselect Oxidative DNA Damage ELISA Kit (Cell Biolabs, STA-320) according to the manufacturer's instructions. In brief, the 96-well plate was coated with 8-oxoG conjugate (1 μg/ml). Total DNA was extracted from inner ear tissues using the DNeasy Blood & Tissue Kit (Qiagen), converted to single-stranded DNA at 95 °C for 5 min, and then immediately cooled on ice. DNA samples were digested to nucleosides by incubating with 5–20 units of nuclease P1 (Sigma-Aldrich) for 2 h at 37 °C in a final concentration of 20 mM sodium acetate, pH 5.2, followed by treatment with 5–10 units of alkaline phosphatase (Sigma-Aldrich) for 1 h at 37 °C in a final concentration of 100 mM Tris, pH 7.5. The reaction mixture was centrifuged for 5 min at 6000 × g and the supernatant was used for the 8-oxoG ELISA. Fifty microliters of samples or 8-oxoG standards were added to the wells of the 8-oxoG conjugate-coated plate and incubated for 10 min at room temperature on an orbital shaker. Fifty microliters of the diluted anti-8-oxoG antibody was added to each well and incubated for 1 h at room temperature on an orbital shaker. After washing with 1x washing buffer three times, 100 μl of the diluted secondary antibody-enzyme conjugate was added to all wells and incubated at room temperature for 1 h on an

orbital shaker. After washing with 1× washing buffer three times, 100 μl of substrate solution was added to each well and incubated for 10 min at room temperature. The reaction was stopped by adding 100 μl of stop solution into each well. The absorbance was read at 450 nm in a spectrometer (Bio-Tek). We used four to five mice per group for assessment of 8-oxoguanine levels.

For the protein carbonyl immunoassay, inner ear tissues were homogenized using a tissue grinder (Wheaton Dounce Tissue Grinder, Fisher Scientific) containing 1 ml of Tris buffer (10 mM Tris, 1 mM EDTA, 320 mM sucrose, pH 7.4) on ice and then centrifuged at 15,000 × g for 10 min at 4 °C. Levels of the oxidative protein damage marker, protein carbonyl, were measured with inner ear whole cell lysate using the OxiSelect Protein Carbonyl ELISA kit (Cell Biolabs STA-310) according to the manufacturer's instructions. In brief, BSA standards and whole cell lysates diluted to 10 μg/ml in 1× PBS were adsorbed onto a 96-well plate overnight at 4 °C. All samples and standards were run in duplicate. After washing three times in 1× PBS, samples were derivatized by adding 100 μl of 1× DNPH solution per well and then were incubated at room temperature for 45 min in the dark. Samples were washed with 1× PBS/Ethanol (1:1, v/v) with incubation for 5 min for a total of 5 times followed by washing in 1× PBS two times. The plate was incubated with blocking buffer (5% skim milk in 1× PBS) for 1 h at room temperature on an orbital shaker. Samples were washed with 1× wash buffer three times. The plate was incubated with the primary antibody anti-DNP (1:1000, diluted in the blocking buffer) for 1 h at room temperature on an orbital shaker. The plate was washed with 1x wash buffer three times. The plate was incubated with the HRP conjugated secondary antibody (1:1000, diluted in the blocking buffer) for 1 h at room temperature on an orbital shaker. After washing with 1× washing buffer five times, 100 μl of substrate solution was added to each well and incubated for 5 min at room temperature on an orbital shaker. The reaction was stopped by adding 100 μl of stop solution into each well. The absorbance was read at 450 nm in a spectrometer (Bio-Tek). Protein carbonyl content was calculated by comparing with a standard curve that is prepared from predetermined reduced and oxidized BSA standards. We used 5 mice per group for assessment of protein carbonyl levels.

**Measurement of total GSH and GSSG**. Labyrinth tissues were homogenized using a tissue grinder (Wheaton Dounce Tissue Grinder, Fisher Scientific) containing 1 ml of lysis buffer (10 mM Tris, 20 mM EDTA, 320 mM sucrose, pH 7.4) on ice and then centrifuged at 12,000 g for 10 min at 4 °C to get a cytosolic lysate (supernatant). One hundred microliters of the cytosolic lysate were used for the measurements of cytosolic glutathione contents. The rates of 5'-thio-2-nitrobenzoic acid (TNB) formation were calculated, and the total glutathione (tGSH) and GSSG concentrations in the samples were determined by using linear regression to calculate the values from the standard curve. The GSH concentration was determined by subtracting the GSSG concentration from the tGSH concentration. All samples were run in duplicate. All reagents used in this assay were purchased from Sigma-Aldrich.

**Statistical analysis**. ANOVA with Bonferroni post hoc testing (GraphPad Prism 4.03) were used to analyze the ABR thresholds, SGN densities, SV thickness, strial capillary numbers, % HC loss, GST activities, and 4-HNE-positive SGNs. Unpaired two-tailed Student's t test was used to analyze wave I amplitudes and latencies, mRNA expression of Gsta4, Nrf2, and genes involved in Phase II detoxification, 8-oxoguanine levels, protein carbonyl levels, GSTA4 protein levels, GSH and GSSG levels, and GSH/GSSG ratios.

**Reporting summary**. Further information on research design is available in the Nature Research Reporting Summary linked to this article.

## Data availability
The authors declare that all data supporting the findings of this study are available within the article (and its Supplementary Information files). The source data underlying Figs. 1a, 2a–e, 3g–n, 4g–p, 5a–d, 6a, b, 7a, b, 8a–c, 9a–c and Supplementary Fig. 1a, 2, 3a, b, 4g, h, 5g, h, 6a, b, 7a, b, and 8a, b are provided as a Source Data file.

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

## Acknowledgements

This research was supported by R03 DC011840 (SS), R01 DC012552 (SS), and R01 DC014437 (SS) from the National Institute of Health and National Institute on Deafness and Communication Disorders, RO1 AG037984 (T.C.F. and A.K.), the Claude D. Pepper Older Americans Independence Centers at the University of Florida (P30 AG028740) from the National Institute of Health and National Institute on Aging, the Evelyn F. McKnight Brain Research Foundation (T.C.F., A.K., and S.S.), and the Japan Society for the Promotion of Science (JSPS) Grant-in-Aid for Scientific Research (A) (Grant number 26253081) and for Scientific Research (S) (Grant number 23228003).

## Author contributions

H.P., M.K., and C.R. performed the research, analyzed the data, designed research, and wrote the paper; E.A.S., A.K., D.D., C.H., K.W., K.B., M.S., Y.K., M.S.T., A.S.G., I.C., and U.B. performed the research; P.J.L., T.M., M.T., T.C.F., and R.S. analyzed the data and wrote the paper; and S.S. designed the research, analyzed the data and wrote the paper.

## Additional information

**Competing interests:** The authors declare no competing interests.

