## [Peer Review File · Nature Communications]

Reviewers' comments:

Reviewer #1 (Remarks to the Author):

This is an interesting manuscript which provides novel information concerning the role of glutathione transferase alpha4 (GSTA4 alpha) in mitigating cisplatin ototoxicity. The novelty of this report is that the investigators have shown that GSTA4 alpha is essential for reducing cisplatin ototoxicity in female mice (but not in male mice). Additionally, the role of GST4A alpha involves detoxification of 4-hydroxynon-2-enal (4-HNE). The strengths of this study is that it highlights the importance of cochlear GSTs in protecting the cochlea against cisplatin/oxidative stress conditions, an area which has been poorly studied. High concentration of GST4A alpha in the spiral ganglion neurons and stria vascularis could explain the toxicity of cisplatin in these regions. A major limitation of this study is that it does not explain why GSTA4 is important in females and why higher levels of basal expression and induction by cisplatin does not confer resistance to cisplatin (compare to males where these features of GSTA4 alpha are not observed).

1. While the loss of SG neurons and thickness of stria vascularis affect all regions of the cochlea (Fig. 3-4), deficits in ABRs are localized to the higher frequencies (Fig. 2). Please comment.
2. While GSTA4 is present in the cochlea of both males and females the lack of discernable response to cisplatin in GSTA4^{-/-} in males is not clear. The authors show that there was a significant increase in 4-HNE in the cochlea in GSTA4^{-/-} female mice. What GST isoforms are involved in detoxifying 4-HNE in males, since their levels were not different in wild type versus GSTA4^{-/-} males? The conclusion made in Fig. 6e is that cisplatin increases GSTA4 protein levels in males and females which was not statistically significant. The question is why was this interpretation of the data even included if the conclusion (increases) was that these changes were not significant. A closer examination of the data indicate that the number of animals used were 3. Since this is an important issue, it would seem that larger group sizes would be used in this and other studies.
3. Figure 7: Again the number of animals used was small (N=3, panel a-c). How is the data presented in panel c normalized for loading?
4. Fig. 7: The divergence between the lack of effect of GSTA 4 knockout in hearing loss and morphological damage to the cochlea (in male mice) as compared to significant increases in accumulation of 8-OHdG (DNA damage marker) in these mice is surprising. Doesn't DNA damage contribute to cell death and hearing loss? This finding would support a role of GSTA 4 alpha in mitigating DNA damage.
5. It would be important to show that elimination of estrogen (by ovariectomy) reduce the benefits afforded by GST4A alpha in female mice. Does ovariectomy reduce constitutive expression of cochlear GSTA4 in females? Does ovariectomy increase susceptibility of female mice to cisplatin? A larger role of estrogen in regulation of GSTA4 alpha is needed.

Reviewer #2 (Remarks to the Author):

The study of Park et al. examines the role of the enzyme GSTA4 in protection against the ototoxicity of the cancer chemotherapeutic agent cisplatin. Hearing loss after cisplatin therapy is a serious comorbidity in cancer patients and is replicated in rodent models. Thus the study is novel and of high potential clinical significance. The data set comparing cisplatin effects in male and female wild type and GSTA4 $-/-$ is comprehensive and includes auditory testing, morphological analysis, immunohistochemical analysis and biochemical analysis. Statistical analysis appears appropriate and the sexual dimorphism data appear convincing and consistent across different measures. The data is interpreted as indicating that GSTA4 is essential for protection against cisplatin ototoxicity only in females even though both sexes express GSTA4 and cisplatin produces similar levels of oxidative stress in both sexes. This interpretation appears to somewhat overstate the data. Ototoxicity appears to still occur in female mice, it is just exacerbated in the GSTA4 $-/-$ genotype accompanied by increases in HNE adducts whereas toxicity in males appears GSTA4 $-/-$ independent. A more accurate interpretation of the results is that the induction of GSTA4 seen only in the female after cisplatin exposure contributes to protection against ototoxicity in that sex consistent with reduction of short chain aldehyde adduction of proteins which have previously been implicated in contributing to cytotoxicity and inflammation in other tissues under other conditions resulting in ROS production. It appears to be the lack of induction in males that is responsible for the lack of a genotype effect. Since GSTA4 is constitutively expressed in both sexes one might expect to see increased injury in both sexes in GSTA4 $-/-$ mice. Since this does not occur, there must also be a GSTA4 independent mechanism of toxicity at play. The discussion should be revised to reflect this.

Reviewers' comments:

Reviewer #1:

1. *General comments: This is an interesting manuscript which provides novel information concerning the role of glutathione transferase alpha4 (GSTA4 alpha) in mitigating cisplatin ototoxicity. The novelty of this report is that the investigators have shown that GSTA4 alpha is essential for reducing cisplatin ototoxicity in female mice (but not in male mice). Additionally, the role of GST4A alpha involves detoxification of 4-hydroxynon-2-enal (4-HNE). The strengths of this study is that it highlights the importance of cochlear GSTs in protecting the cochlea against cisplatin/oxidative stress conditions, an area which has been poorly studied. High concentration of GST4A alpha in the spiral ganglion neurons and stria vascularis could explain the toxicity of cisplatin in these regions. A major limitation of this study is that it does not explain why GSTA4 is important in females and why higher levels of basal expression and induction by cisplatin does not confer resistance to cisplatin (compare to males where these features of GSTA4 alpha are not observed).*

Response: Please see our response to Reviewer 1's comment #7 below.

2. *While the loss of SG neurons and thickness of stria vascularis affect all regions of the cochlea (Fig. 3-4), deficits in ABRs are localized to the higher frequencies (Fig. 2). Please comment.*

Response: In general, ABR thresholds were higher in the low frequencies (4-8 kHz) and high frequencies (32-64 kHz) in both males and females after cisplatin treatment.

3. *While GSTA4 is present in the cochlea of both males and females the lack of discernable response to cisplatin in GSTA4^{-/-} in males is not clear. The authors show that there was a significant increase in 4-HNE in the cochlea in GSTA4^{-/-} female mice. What GST isoforms are involved in detoxifying 4-HNE in males, since their levels were not different in wild type versus GSTA4^{-/-} males?*

Response: Previous studies have shown that GSTA1, GSTA2, GSTA4, GSTM1 and GSTT1 enzymes possess antioxidant activity towards lipid hydroperoxides, including 4-HNE, in both males and females (Henderson et al., Drug Metab Rev. 2011 May;43(2):152-64. PMID: 21425933; Simic et al., Nat Rev Urol. 2009 May;6(5):281-9. PMID: 19424176; Coles et al., Methods Enzymol. 2005;401:9-42. PMID: 16399377; Simic et al., Amino Acids. 2006 Jun;30(4):495-8. PMID: 16773246). This statement was added on page 16, in line 344 in the revised manuscript.

4. *The conclusion made in Fig. 6e is that cisplatin increases GSTA4 protein levels in males and females which was not statistically significant. The question is why was this interpretation of the data even included if the conclusion (increases) was that these changes were not significant. A closer examination of the data indicate that the number of animals used were 3. Since this is an important issue, it would seem that larger group sizes would be used in this and other studies.*

Response: We performed gene expression analysis of inner ear tissues from young male, female, and ovariectomized female (OVX) CBA/CaJ mice using pathway-focused-PCR arrays containing 84 genes involved in Phase II detoxification (n=5). We found that female CBA/CaJ mice displayed higher mRNA expression of 9 genes involved in Phase II detoxification, including *Gsta4*, compared to males (new Fig. 7). In contrast, ovariectomy downregulated 25 genes involved in Phase II detoxification, including *Gsta4* (new Fig. 8). We also re-measured GSTA4 protein levels in the inner ear tissues from the same groups of CBA/CaJ mice by capillary electrophoresis-based immunoassay using a Wes system (n=5). Although there were no differences in GSTA4 protein levels between male and female CBA/CaJ mice, OVX mice showed significantly lower GSTA4 protein levels compared to female mice (new Fig. 9). These results show that the constitutive gene expression of *Gsta4* in the inner ear of female mice is higher than that of male mice.

We note that performing protein/DNA/RNA analysis using mouse inner ear tissues is a challenge because of the small amount of inner ear proteins/DNA/RNA available from an individual mouse. This

is why the number of animals used for the physiological test (ABR testing) was 9-10, while the number of animals used for the protein/biochemical/RNA/DNA analyses was 3-5.

5. *Figure 7: Again the number of animals used was small (N=3, panel a-c). How is the data presented in panel c normalized for loading?*

Response: The number of animals used in the assessment of 8-oxoG levels was 5, but not 3 (n=5). We apologize about this error. Please see new Fig. 11a and figure legend. To further investigate whether ovariectomy affects cisplatin-induced oxidative DNA damage in female *Gsta4^{-/-}* mice, we measured 8-oxoG levels in the inner ears from cisplatin-treated female, male and OVX *Gsta4^{-/-}* mice using the Oxidative DNA Damage ELISA kit (n=4-5). There were no differences in 8-oxoG levels between male and female *Gsta4^{-/-}* mice or female and OVX *Gsta4^{-/-}* mice (new Fig. 11b-c).

We also measured protein carbonyl levels in the inner ears from cisplatin-treated female, male and OVX *Gsta4^{-/-}* mice using the Protein Carbonyl ELISA kit (n=5). There were also no differences in levels of protein carbonyl, between male and female *Gsta4^{-/-}* mice or female and OVX *Gsta4^{-/-}* mice (new Supplementary Fig. 7a-b).

As for old Fig. 7c, the signal intensity of the molecular weight standard was used as a control for the experimental samples to measure protein carbonyl levels using the Oxyblot Protein Oxidation Detection kit (EMD Millipore, Billerica, MA) according to the manufacturer's instructions. We note that old Fig. 7b-c was replaced by new Supplementary Fig. 7a-b.

6. *Fig. 7: The divergence between the lack of effect of GSTA 4 knockout in hearing loss and morphological damage to the cochlea (in male mice) as compared to significant increases in accumulation of 8-OHdG (DNA damage marker) in these mice is surprising. Doesn't DNA damage contribute to cell death and hearing loss? This finding would support a role of GSTA 4 alpha in mitigating DNA damage.*

Response: We agree that this finding suggests that GSTA4 also plays a role in reducing cisplatin-induced oxidative DNA damage in the inner ear of both male and female mice. We speculate that GSTA4 can mediate reduction of cisplatin-induced oxidative DNA damage because 4-HNE is known to covalently modify DNA and GSTA4 has high catalytic efficiency toward 4-HNE (Balogh et al. *Drug Metab Rev* 2011, PMID: 21401344; Hubatsch et al., *Biochem J* 1998, PMID: 9461507; Zimniak et al. *J Biol Chem* 1994, PMID: 7904605). This discussion was added on page 16, line 350 of the revised manuscript.

7. *It would be important to show that elimination of estrogen (by ovariectomy) reduce the benefits afforded by GST4A alpha in female mice. Does ovariectomy reduce constitutive expression of cochlear GSTA4 in females? Does ovariectomy increase susceptibility of female mice to cisplatin? A larger role of estrogen in regulation of GSTA4 alpha is needed.*

Response: We have performed ovariectomy in both female CBA/CaJ and *Gsta4^{-/-}* mice. We then performed gene expression analysis of inner ear tissues from young male, female, and ovariectomized female (OVX) CBA/CaJ mice using PCR arrays containing 84 genes involved in Phase II detoxification (n=5). We also re-measured GSTA4 protein levels in the inner ear tissues from the same groups of CBA/CaJ mice by capillary electrophoresis-based immunoassay using a Wes system (n=5). These results show that: 1) the constitutive gene expression of *Gsta4* in the inner ear of female mice is higher than that of male mice and 2) ovariectomy reduces the expression levels of both GSTA4 protein and *Gsta4* mRNA in the inner ears.

Moreover, we measured 8-oxoG levels in the inner ears from cisplatin-treated female, male and OVX *Gsta4^{-/-}* mice. We also measured protein carbonyl levels in the inner ears from cisplatin-treated female, male and OVX *Gsta4^{-/-}* mice. There were no differences in 8-oxoG levels or protein carbonyl levels

between male and female *Gsta4*^{-/-} mice or female and OVX *Gsta4*^{-/-} mice. These results show that ovariectomy does not increase susceptibility to cisplatin-induced oxidative DNA or protein damage in female *Gsta4*^{-/-} mice.

Thank you for your highly insightful comments and suggestions. We believe your comments enabled us to greatly improve the quality of our manuscript.

Reviewer #2:

1. *The study of Park et al. examines the role of the enzyme GSTA4 in protection against the ototoxicity of the cancer chemotherapeutic agent cisplatin. Hearing loss after cisplatin therapy is a serious comorbidity in cancer patients and is replicated in rodent models. Thus the study is novel and of high potential clinical significance. The data set comparing cisplatin effects in male and female wild type and GSTA4 -/- is comprehensive and includes auditory testing, morphological analysis, immunohistochemical analysis and biochemical analysis. Statistical analysis appears appropriate and the sexual dimorphism data appear convincing and consistent across different measures.*

The data is interpreted as indicating that GSTA4 is essential for protection against cisplatin ototoxicity only in females even though both sexes express GSTA4 and cisplatin produces similar levels of oxidative stress in both sexes. This interpretation appears to somewhat overstate the data. Ototoxicity appears to still occur in female mice, it is just exacerbated in the GSTA4 -/- genotype accompanied by increases in HNE adducts whereas toxicity in males appears GSTA4 -/- independent. A more accurate interpretation of the results is that the induction of GSTA4 seen only in the female after cisplatin exposure contributes to protection against ototoxicity in that sex consistent with reduction of short chain aldehyde adduction of proteins which have previously been implicated in contributing to cytotoxicity and inflammation in other tissues under other conditions resulting in ROS production. It appears to be the lack of induction in males that is responsible for the lack of a genotype effect. Since GSTA4 is constitutively expressed in both sexes one might expect to see increased injury in both sexes in GSTA4 -/- mice. Since this does not occur, there must also be a GSTA4 independent mechanism of toxicity at play. The discussion should be revised to reflect this.

Response: We agree that the induction of GSTA4 by cisplatin contributes to protection against ototoxicity in females, whereas cisplatin ototoxicity in males appear *Gsta4*^{-/-} independent. We have revised the abstract, result and discussion sections to reflect this point.

We performed gene expression analysis of inner ear tissues from young male, female, and ovariectomized female (OVX) CBA/CaJ mice using pathway-focused-PCR arrays containing 84 genes involved in Phase II detoxification (n=5). We found that female CBA/CaJ mice displayed higher mRNA expression of 9 genes involved in Phase II detoxification, including *Gsta4*, *Gstm2*, and *Gstt1*, compared to males (new Fig. 7). In contrast, ovariectomy downregulated 25 genes involved in Phase II detoxification, including *Gsta4*, *Gstk1*, *Gstm2*, *Gstm4*, *Gstp1*, *Gstt1*, and *Gstt2* (new Fig. 8). Thus, we speculate that cisplatin stimulates GSTA4 activity in the inner ear of females, but not males, likely because the constitutive expression of inner ear *Gsta4* is significantly higher in female than males, and/or females have higher levels of estrogen.

Previous studies have shown that GSTA1, GSTA2, GSTA4, GSTM1 and GSTT1 also possess antioxidant activity towards lipid hydroperoxides, including 4-HNE. Therefore, we speculate that a complete loss of *Gsta4* leads to more pronounced cisplatin ototoxicity in females compared to males likely because: 1) females use GSTA4 as a primary detoxification enzyme to remove 4-HNE due to the higher abundance of *Gsta4*, 2) GSTA4 has higher catalytic efficiency toward 4-HNE than other GSTs, and/or 3) males do not use GSTA4 as a primary detoxification enzyme to remove 4-HNE due to the lower abundance of *Gsta4*. These statements were added on page 16, line 342 of the revised manuscript.

Thank you for your highly insightful comments.

REVIEWERS' COMMENTS:

Reviewer #1 (Remarks to the Author):

This manuscript is acceptable for publication.

Reviewer #2 (Remarks to the Author):

This revised manuscript from Park et al. on the role of GSTA4 in protection from cisplatin ototoxicity in females has responded positively to previous reviewer issues. Additional data is presented on sexual dimorphism of GSTA4 and other phase II enzymes in the inner ear and response of female phase II enzyme expression to ovariectomy. In addition data are presented on sexual dimorphism of Nrf2 which regulates GST expression and the substrate specificity of other GSTs toward HNE is now also added to the discussion. As a result the paper is much improved.

Reviewers' comments:

Reviewer #1:

1. *This manuscript is acceptable for publication.*

Response: Thank you for your highly insightful comments and suggestions.

Reviewer #2:

1. *This revised manuscript from Park et al. on the role of GSTA4 in protection from cisplatin ototoxicity in females has responded positively to previous reviewer issues. Additional data is presented on sexual dimorphism of GSTA4 and other phase II enzymes in the inner ear and response of female phase II enzyme expression to ovariectomy. In addition data are presented on sexual dimorphism of Nrf2 which regulates GST expression and the substrate specificity of other GSTs toward HNE is now also added to the discussion. As a result the paper is much improved.*

Response: Thank you for your highly insightful comments and suggestions. We believe your comments enabled us to greatly improve the quality of our manuscript.